# Eukaryotic initiation factor EIF-3.G augments mRNA translation efficiency to regulate neuronal activity

**Stephen M Blazie, Seika Takayanagi-Kiya, Katherine A McCulloch, Yishi Jin\***

Section of Neurobiology, Division of Biological Sciences, University of California San Diego, La Jolla, United States

**Abstract** The translation initiation complex eIF3 imparts specialized functions to regulate protein expression. However, understanding of eIF3 activities in neurons remains limited despite widespread dysregulation of eIF3 subunits in neurological disorders. Here, we report a selective role of the *C. elegans* RNA-binding subunit EIF-3.G in shaping the neuronal protein landscape. We identify a missense mutation in the conserved Zinc-Finger (ZF) of EIF-3.G that acts in a gain-of-function manner to dampen neuronal hyperexcitation. Using neuron-type-specific seCLIP, we systematically mapped EIF-3.G-mRNA interactions and identified EIF-3.G occupancy on GC-rich 5′UTRs of a select set of mRNAs enriched in activity-dependent functions. We demonstrate that the ZF mutation in EIF-3.G alters translation in a 5′UTR-dependent manner. Our study reveals an in vivo mechanism for eIF3 in governing neuronal protein levels to control neuronal activity states and offers insights into how eIF3 dysregulation contributes to neurological disorders.

## Introduction

Protein synthesis is principally regulated by variations in the translation initiation mechanism, whereby multiple eukaryotic initiation factors (eIF1 through 6) engage elongation-competent ribosomes with the mRNA open reading frame (*Sonenberg and Hinnebusch, 2009*). eIF3 is the largest translation initiation complex, composed of 13 subunits in metazoans, with versatile functions throughout the general translation initiation pathway (*Valášek et al., 2017*). Extensive biochemical and structural studies have shown that eIF3 promotes translation initiation by orchestrating effective interactions between the ribosome, target mRNA, and other eIFs (*Smith et al., 2016*; *Cate, 2017*). Mutations and misexpression of various subunits of eIF3 are associated with human diseases, such as cancers and neurological disorders (*Gomes-Duarte et al., 2018*), raising the importance to advance mechanistic understanding of eIF3's function in vivo.

Recent work has begun to reveal that different eIF3 subunits can selectively regulate translation in a manner depending on cell type, mRNA targets, and post-translational modification. Interaction of eIF3 RNA-binding subunits with specific 5′UTR stem-loop structures of mRNAs can trigger a translational switch for cell proliferation in human 293 T cells (*Lee et al., 2015*), and can also act as a translational repressor, such as the case for human Ferritin mRNA (*Pulos-Holmes et al., 2019*). Under cellular stress, such as heat shock, the eIF3 complex circumvents cap-dependent protein translation initiation and recruits ribosomes directly to m6A marks within the 5′UTR of mRNAs encoding stress response proteins (*Meyer et al., 2015*). Other specialized translation mechanisms appear to involve activities of particular eIF3 subunits that were previously hidden from view. For example, human eIF3d possesses a cryptic mRNA cap-binding function that is activated by phosphorylation and stimulates pre-initiation complex assembly on specific transcripts (*Lee et al., 2016*; *Lamper et al., 2020*), while eIF3e specifically regulates metabolic mRNA translation (*Shah et al.,*

**\*For correspondence:**
yijin@ucsd.edu

**Competing interests:** The authors declare that no competing interests exist.

*2016*). These findings hint that many other eIF3-guided mechanisms of cell-specific translational control await discovery.

In the nervous system, emerging evidence suggests that eIF3 subunits may have critical functions. Knockdown of multiple eIF3 subunits impairs expression of dendrite pruning factors in developing sensory neurons of *Drosophila* (*Rode et al., 2018*). In mouse brain, eIF3h directly interacts with collybistin, a conserved neuronal Rho-GEF protein underlying X-linked intellectual disability with epilepsy (*Sertie et al., 2010*; *Machado et al., 2016*). In humans, altered expression of the eIF3 complex in the substantia nigra and frontal cortex correlates with Parkinson's Disease progression (*Garcia-Esparcia et al., 2015*). Downregulation of mRNAs encoding eIF3 subunits is observed in a subset of motor neurons in amyotrophic lateral sclerosis patients (*Cox et al., 2010*). Furthermore, a single-nucleotide polymorphism located in the intron of human eIF3g elevates its mRNA levels and is associated with narcolepsy (*Holm et al., 2015*). While these data suggest that eIF3 function in neurons is crucial, mechanistic understanding will require experimental models enabling in vivo investigation of how eIF3 affects protein translation with neuron-type specificity.

Protein translation in *C. elegans* employs all conserved translation initiation factors. We have investigated the mechanisms of protein translation in response to neuronal overexcitation using a gain-of-function (*gf*) ion channel that arises from a missense mutation in the pore-lining domain of the acetylcholine receptor subunit ACR-2 (*Jospin et al., 2009*). The cholinergic motor neurons (ACh-MNs) in the ventral cord of *acr-2(gf)* mutants experience constitutive excitatory inputs, which gradually diminish pre-synaptic strength and cause animals to display spontaneous seizure-like convulsions and uncoordinated locomotion (*Jospin et al., 2009*; *Zhou et al., 2017*). *acr-2(gf)* induces activity-dependent transcriptome changes (*McCulloch et al., 2020*). However, it is unclear how protein translation conducts the activity-dependent proteome changes that sustain function of these neurons.

Here, we demonstrate that *C. elegans* EIF-3.G/eIF3g regulates the translation efficiency of select mRNAs in ACh-MNs. We characterized a mutation (C130Y) in the zinc-finger of EIF-3.G that suppresses behavioral deficits of *acr-2(gf)* without disrupting general protein translation. By systematic profiling of EIF-3.G and mRNA interactions in ACh-MNs, we identified preferential binding of EIF-3.G to long and GC-rich 5'UTRs of mRNAs, many of which encode modulators of ACh-MN activity. We further provided in vivo evidence that EIF-3.G regulates the expression of two of its mRNA targets dependent on their 5'UTRs. Our findings illustrate the selectivity of EIF-3.G in augmenting mRNA translation to mediate neuronal activity changes.

## Results

### A missense mutation in EIF-3.G ameliorates convulsion behaviors caused by cholinergic hyperexcitation

We previously characterized numerous mutations that suppress convulsion and locomotion behaviors of *acr-2(gf)* animals (*McCulloch et al., 2017*). One such suppressor mutation, *ju807*, was found to contain a single nucleotide alteration in *eif-3.G*, encoding subunit G of the EIF-3 complex (*Figure 1A*; see Materials and methods). *C. elegans* EIF-3.G is composed of 262 amino acids, sharing overall 35% or 32% sequence identity with human eIF3g and *S. cerevisiae* TIF35 orthologs, respectively (*Figure 1—figure supplement 1A*). Both biochemical and structural data show that eIF3g/TIF35 proteins bind eIF3i/TIF34 through a domain in the N-terminus (*Figure 1B*; *Valášek et al., 2017*). eIF3g/EIF-3.G also has a predicted CCHC zinc finger followed by an RNA recognition motif (RRM) at the C-terminus (*Figure 1B* and *Figure 1—figure supplement 1A*). The *ju807* mutation changes the second cysteine of the CCHC motif (Cys130, corresponding to Cys160 in human eIF3g) to tyrosine (*Figure 1B*). Hereafter, we designate *eif-3.G(ju807)* as *eif-3.G(C130Y)*.

Compared to *acr-2(gf)* single mutants, *eif-3.G(C130Y); acr-2(gf)* animals exhibited nearly wild-type movement and strongly attenuated convulsion behavior (*Figure 1C*; *Videos 1–3*). *acr-2(gf)* animals carrying heterozygous e*if-3.G(C130Y/+)* showed partial suppression of convulsions (*Figure 1C*). Overexpression of wild type *eif-3.G* full-length genomic DNA in *eif-3.G(C130Y); acr-2(gf)* double mutants restored convulsions to levels similar to *eif-3.G(C130Y/+); acr-2(gf)* (*Figure 1D–E*; Materials and methods). Overexpression of *eif-3.G(C130Y)* full-length genomic DNA in *acr-2(gf)* single mutants also partially suppressed convulsions (*Figure 1C–D*). In wild-type animals,

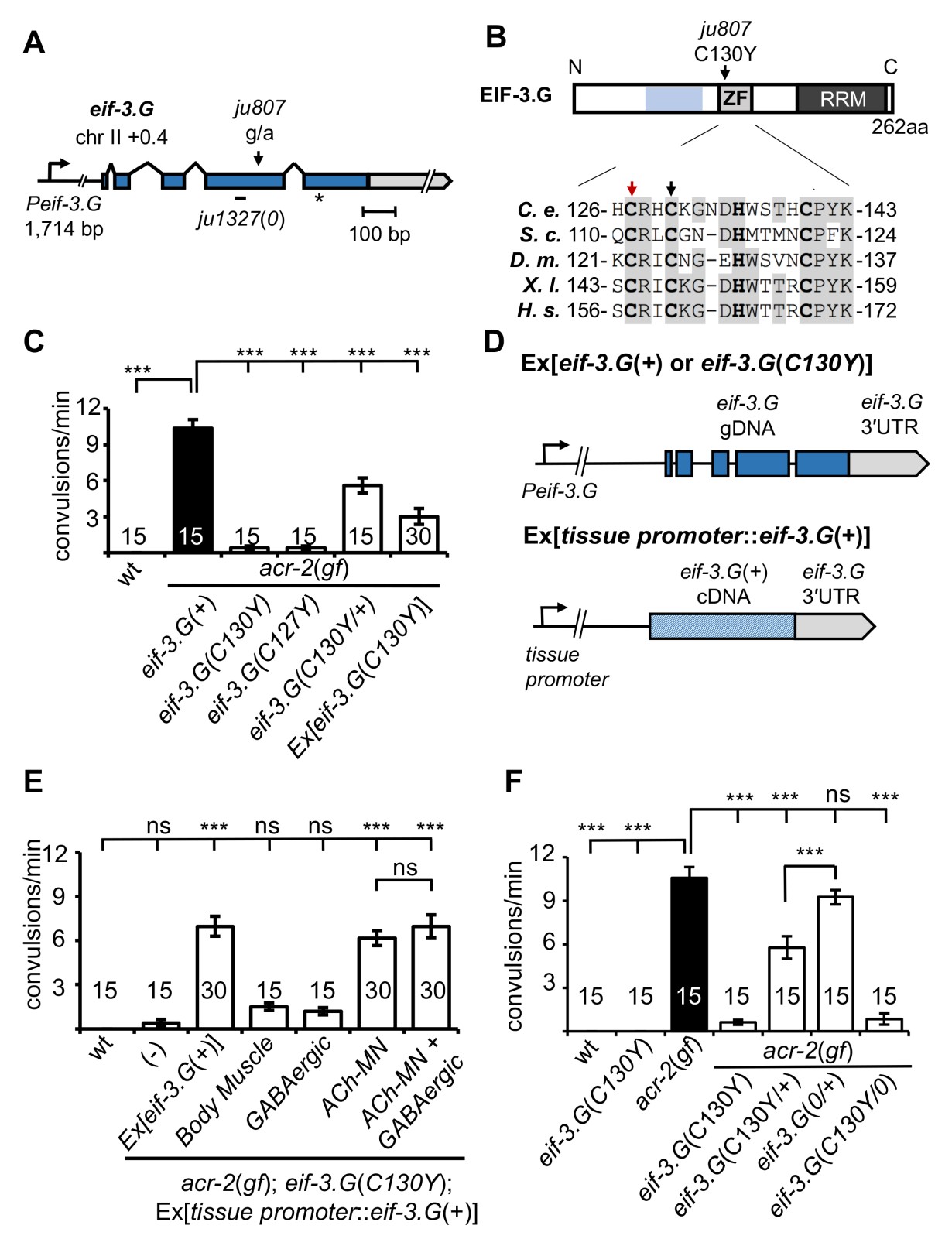

**Figure 1.** *eif-3.G*(C130Y) suppresses *acr-2*(*gf*) convulsion behavior in the cholinergic motor neurons. (**A**) Illustration of the genomic locus of *eif-3.G*: *Peif-3.G* denotes the promoter, blue boxes are exons for coding sequences and gray for 3′UTR. Arrowhead indicates guanine to adenine change in *ju807*; and short line below represents a 19 bp deletion in *ju1327*, designated *eif-3.G(0)*, that would shift the reading frame at aa109, resulting in a premature stop (asterisk) after addition of 84aa of no known homology. (**B**) Illustration of EIF-3.G: shaded blue represents EIF-3.I binding region, ZF for Zinc Finger,

*Figure 1 continued on next page*

*Figure 1 continued*

RRM for RNA Recognition Motif. Below is a multi-species alignment of the zinc finger domain with bold residues as the CCHC motif and gray for conserved residues. *ju807* causes a C130Y substitution (black arrow). C127Y (red arrow, *ju1840*) was generated with CRISPR editing. *C. elegans* (*C. e.*; NP_001263666.1), *S. cerevisiae* (*S. c.*; NP_010717.1), *D. melanogaster* (*D. m.*; NP_570011.1), *X. laevis* (*X. l.*; NP_001087888.1), and *H. sapiens* (*H.s.*; AAC78728.1). (**C**) Quantification of convulsion frequencies of animals of indicated genotypes, with the strains (left to right) as: N2, MT6241, CZ21759, CZ28495, CZ21759, CZ22977. Ex[*eif-3.G(C130Y)*] transgenes (*juEx7015/juEx7016*) expressed full-length genomic DNA cloned from *eif-3.G(ju807)*. (**D**) Illustration of *eif-3.G* expression constructs: top shows the transgene expressing genomic *eif-3.G*(+ for wild type and *C130Y* for *ju807*) with the endogenous *eif-3.G* promoter and 3'UTR, and coding exons in blue; bottom shows cell-type expression of *eif-3.G* cDNA driven by tissue-specific promoters (*Pmyo-3*- body muscle, *Punc-25*- GABAergic motor neurons, *Punc-17β* - cholinergic motor neurons). (**E**) Quantification of convulsion frequencies shows that convulsion behavior of *eif-3.G(C130Y)*; *acr-2(gf)* double mutants is rescued by transgenes that overexpress *eif-3.G*(+) genomic DNA or an *eif-3.G*(+) cDNA in the ACh-MNs, but not in the GABAergic motor neurons or body muscle. Strains (left to right)- N2, CZ21759, CZ23125/ CZ23126, CZ22980/ CZ22981, CZ23791/ CZ23880, CZ22982/ CZ22983, CZ27881/ CZ27882. (**F**) Quantification of convulsion frequencies in animals of the indicated genotypes (left to right)- N2, CZ22917, MT6241, CZ21759, CZ28495, CZ21759, CZ21759, CZ23310, CZ26828. Data in (**D-F**) are shown as mean ± SEM and sample size is indicated within or above each bar. Statistics: (***) p<0.001, (ns) not significant by one-way ANOVA with Bonferroni's post hoc test.

The online version of this article includes the following source data and figure supplement(s) for figure 1:

**Source data 1.** Source data for *Figure 1C*.
**Source data 2.** Source data for *Figure 1E*.
**Source data 3.** Source data for *Figure 1F*.
**Figure supplement 1.** EIF-3.G is highly conserved and expressed ubiquitously.
**Figure supplement 2.** Motor neuron development is normal in *eif-3.G(C130Y)* animals.
**Figure supplement 2—source data 1.** Source data for *Figure 1—figure supplement 2A*.
**Figure supplement 2—source data 2.** Source data for *Figure 1—figure supplement 2B*.
**Figure supplement 3.** EIF-3.G(C130Y) modulation of convulsion behavior does not involve reduced EIF-3 complex dosage or ACR-2 expression.
**Figure supplement 3—source data 1.** Source data for *Figure 1—figure supplement 3A*.
**Figure supplement 3—source data 2.** Source data for *Figure 1—figure supplement 3B*.

overexpression of *eif-3.G*(+) or *eif-3.G(C130Y)* caused no observable effects on locomotion. These data show that *eif-3.G(C130Y)* acts in a semi-dominant manner to ameliorate convulsion and uncoordinated locomotion behaviors of *acr-2(gf)*. To further test that altering the EIF-3.G zinc finger motif accounts for the observed suppression of *acr-2(gf)*, we edited the first cysteine of the CCHC motif (Cys127) to tyrosine using CRISPR-Cas9, and found that *eif-3.G(C127Y)* suppressed *acr-2(gf)* convulsions to levels identical to *eif-3.G(C130Y)* (*Figure 1B–C*). This data provides support for the importance of the EIF-3.G zinc finger in regulation of ACh-MN activity. Hereafter, we focused our analysis on *eif-3.G(C130Y)*.

We next determined in which cell types *eif-3.G(C130Y)* functions using cell-specific expression analysis (*Figure 1D*; also see Materials and methods). *acr-2(gf)* phenotypes arise from a hyperactive ACR-2-containing ion channel expressed in the ventral cord cholinergic motor neurons (ACh-MNs) (*Jospin et al., 2009*). We found that overexpressing *eif-3.G*(+) cDNA in ACh-MNs (*Punc-17β*) restored convulsions of *eif-3.G(C130Y)*; *acr-2(gf)* animals to a similar degree as those expressing full-length *eif-3.G*(+) under the endogenous promoter (*Peif-3.G*) (*Figure 1E*). In contrast, overexpression of *eif-3.G*(+) cDNA in either ventral cord GABAergic neurons (GABA-MNs, *Punc-25*) or body muscle (*Pmyo-3*) in *eif-3.G(C130Y)*; *acr-2(gf)* animals caused no detectable effects (*Figure 1E*). Co-expression of *eif-3.G*(+) in both ACh-MNs and GABA-MNs showed similar effects on *eif-3.G(C130Y)*; *acr-2(gf)* animals to that from expressing *eif-3.G*(+) in ACh-MNs alone (*Figure 1E*). Thus, *eif-3.G(C130Y)* functions in ACh-MN to modulate *acr-2(gf)* behaviors.

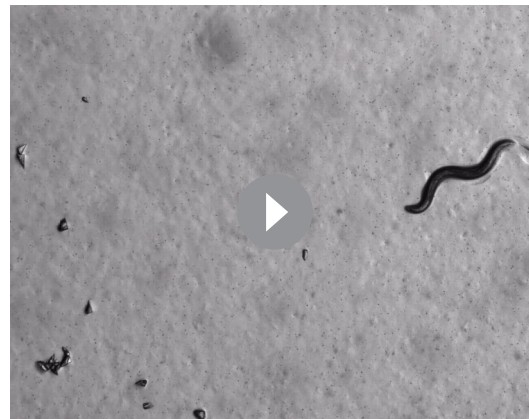

**Video 1.** N2 [Wild type] *C. elegans* movement on solid nematode growth media.
https://elifesciences.org/articles/68336#video1

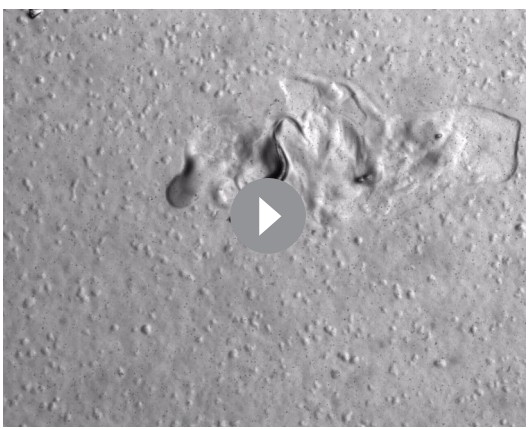

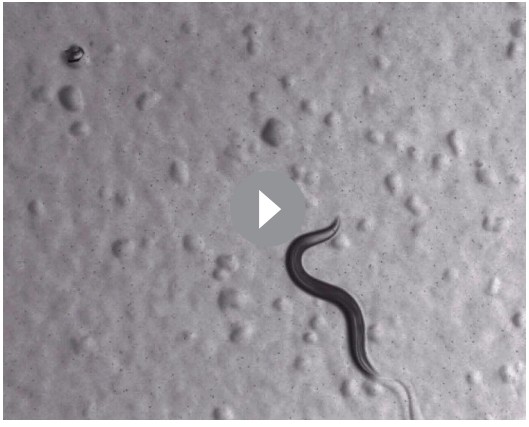

**Video 2.** MT6241 [*acr-2*(*gf*)] *C. elegans* movement on solid nematode growth media.
https://elifesciences.org/articles/68336#video2

**Video 3.** CZ21759 [*eif-3.G*(*C130Y*); *acr-2*(*gf*)] *C. elegans* movement on solid nematode growth media.
https://elifesciences.org/articles/68336#video3

## EIF-3.G(C130Y) selectively affects translation in ACh-MNs

*eif-3.G*(*C130Y*) single mutants exhibit normal development, locomotion, and other behaviors (such as male mating and egg-laying) indistinguishably from wild-type animals (**Figure 1F**, **Video 4**). Axon morphology and synapse number of ACh-MNs were also normal in *eif-3.G*(*C130Y*) animals (**Figure 1—figure supplement 2A–B**). To dissect how the C130Y mutation affects EIF-3.G function, we next generated a genetic null mutation (*ju1327*) using CRISPR editing (**Figure 1A** and **Figure 1—figure supplement 1A**; designated *eif-3.G*(*0*), see Materials and methods). Homozygous *eif-3.G*(*0*) animals arrested development at L1 stage, consistent with EIF-3 complex members being required for *C. elegans* development (**Kamath et al., 2003**). *eif-3.G*(*0/+*); *acr-2*(*gf*) animals were indistinguishable from *acr-2*(*gf*) single mutants (**Figure 1F**). We additionally tested null mutations in EIF-3.E and EIF-3.H, two essential subunits of EIF-3 complex, and found that *acr-2*(*gf*) animals carrying heterozygous null mutations in either *eif-3* subunit gene showed convulsions similar to *eif-3.G*(*0/+*); *acr-2*(*gf*) (**Figure 1—figure supplement 3A**). Moreover, hemizygous *eif-3.G*(*C130Y/0*) animals are healthy at all stages and suppress behaviors of *acr-2*(*gf*) to levels comparable to *eif-3.G*(*C130Y*) (**Figure 1F**). Reducing one copy of *eif-3.H*(*+*) or *eif-3.E*(*+*) in *eif-3.G*(*C130Y*); *acr-2*(*gf*) animals also did not modify the suppression effect of *eif-3.G*(*C130Y*) (**Figure 1—figure supplement 3A**). These observations suggest that *eif-3.G*(*C130Y*) retains sufficient function of wild-type *eif-3.G*, and likely affects a regulatory activity that is not dependent on EIF-3 subunit dosage.

We considered that EIF-3.G(C130Y) could alter EIF-3.G protein levels in ACh-MNs. To test this, we generated single-copy chromosomal integrated transgenes expressing EIF-3.G(WT) or EIF-3.G(C130Y) tagged with GFP at the N-terminus under the control of the endogenous *eif-3.G* promoter (Materials and methods and **Supplementary file 1**). Fluorescence from both GFP::EIF-3.G(WT) and GFP::EIF-3.G(C130Y) was observed in all somatic cells (**Figure 1—figure supplement 1B**). In ACh-MNs, both proteins showed cytoplasmic localization (**Figure 2B**). The GFP::EIF-3.G(WT) transgene rescued *eif-3.G*(*0*) to adults (**Supplementary file 1**) and also restored convulsion behavior in the *eif-3.G*(*C130Y*); *acr-2*(*gf*) background (**Figure 2A**). In contrast, the GFP::EIF-3.G(C130Y) transgene reduced convulsion behavior in the *acr-2*(*gf*) background. Furthermore, we introduced the

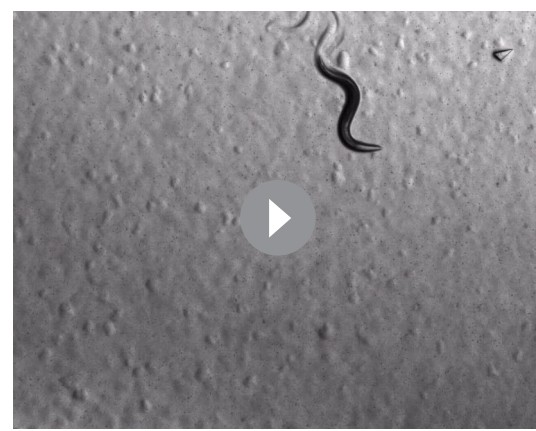

**Video 4.** CZ22197 [*eif-3.G*(*C130Y*)] *C. elegans* movement on solid nematode growth media.
https://elifesciences.org/articles/68336#video4

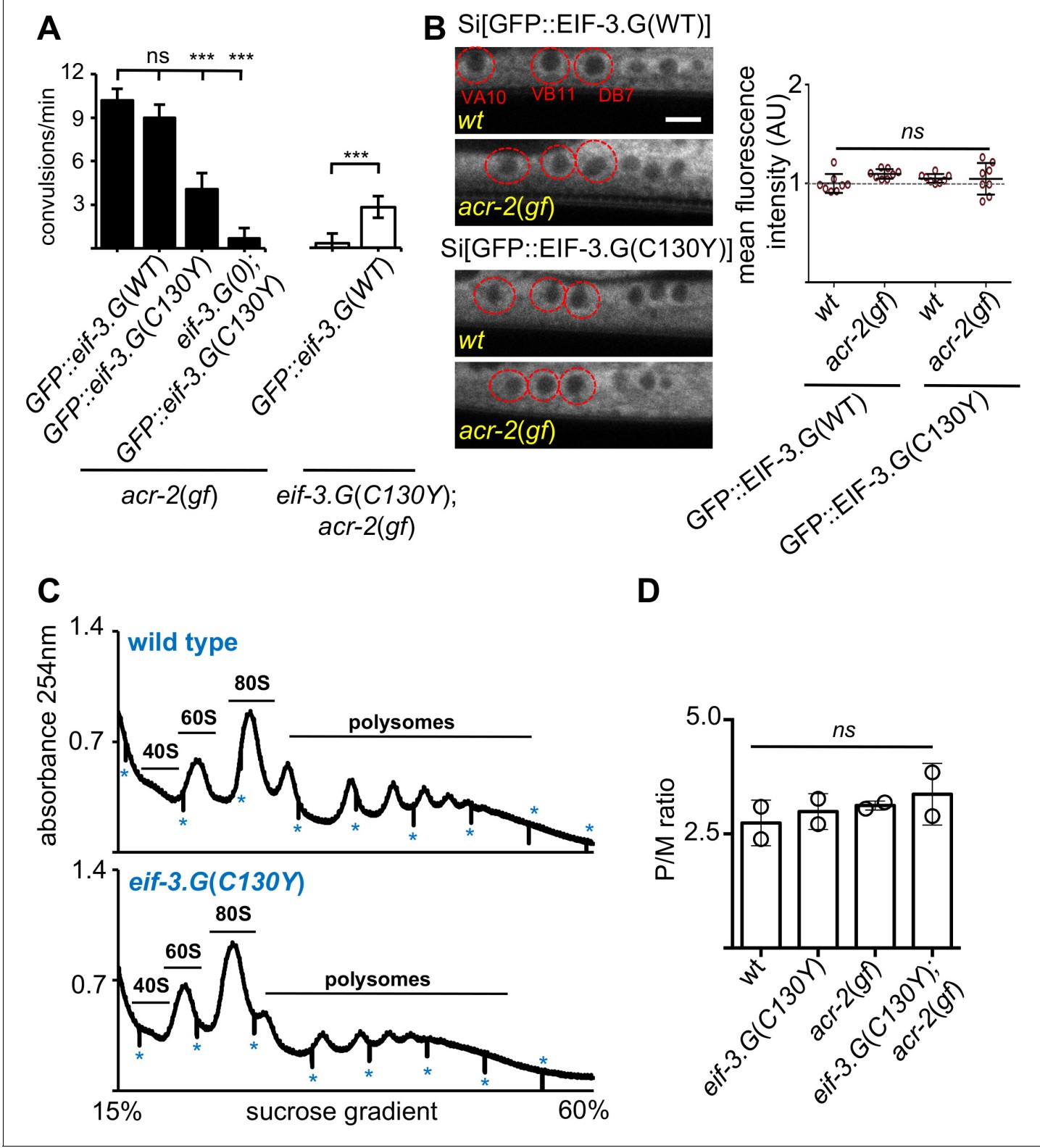

**Figure 2.** *eif-3.G(C130Y)* involves a selective function of EIF-3.G on translational control. (**A**) Quantification of convulsion frequency in animals expressing GFP::EIF-3.G(WT) or GFP::EIF-3.G(C130Y) under *Peif-3.G* in the indicated genetic backgrounds; and the strains (left to right) are: MT6241, CZ24729, CZ24652, CZ28497, CZ21759, CZ28107. Error bars represent ± SEM with n = 15 per sample. (***) P< 0.001, (ns) not significant, by one-way ANOVA with Bonferroni's post-hoc test. (**B**) EIF-3.G(WT) and EIF-3.G(C130Y) show comparable expression in ACh-MNs. Left are representative single-

*Figure 2 continued on next page*

*Figure 2 continued*

plane confocal images of EIF-3.G(WT)::GFP or EIF-3.G(C130Y)::GFP driven by the *Peif-3.G* promoter as single-copy transgenes in L4 animals (head to the left). Red circles mark the soma of VA10, VB11, and DB7 ACh-MN, based on co-expressing a *Pacr-2-mcherry* marker. Scale bar = 4 μm. Right: Mean GFP fluorescence intensities (AU) in ACh-MN soma in animals of the indicated genotypes (n = 8). Each data point represents the mean intensity from VA10, VB11, and DB7 neurons in the same animal and normalized to the mean GFP::EIF-3.G intensity in a wildtype background. Error bars represent ± SEM; (ns) not significant by one-way ANOVA with Sidak's multiple comparisons test. (**C**) Representative polysome profile traces from total mRNA-protein extracts of wild type and *eif-3.G(C130Y)* single mutant animals. Vertical lines (marked by *) within traces indicate the boundaries of fraction collection. (**D**) Polysome::monosome (P/M) ratios calculated based on the area under the respective curves for polysomal and monosome (80S) fractions using two replicates of polysome profiles from total extracts of indicated genotypes. (ns) not significant by one-way ANOVA with Bonferroni's post-hoc test.

The online version of this article includes the following source data for figure 2:

**Source data 1.** Source data for *Figure 2A*.
**Source data 2.** Source data for *Figure 2B*.
**Source data 3.** Source data for *Figure 2C*.

GFP::EIF-3.G(C130Y) transgene into the *eif-3.G(0)*; *acr-2(gf)* background and observed that this transgene rescued the arrested larvae to adults and nearly abolished convulsion behavior (*Figure 2A*). This analysis shows that GFP::EIF-3.G(WT) and GFP::EIF-3.G(C130Y) retain function and lends further support that *eif-3.g(C130Y)* is responsible for the suppression of *acr-2(gf)*. Quantification of GFP levels in the ACh-MNs showed equivalent intensity and localization of GFP::EIF-3.G (WT and C130Y) between wild type and *acr-2(gf)* animals (*Figure 2B*), indicating that EIF-3.G(C130Y) does not increase EIF-3.G protein stability.

We further assessed whether *eif-3.G(C130Y)* alters global translation by performing polysome profile analysis using whole *C. elegans* lysates of L4 stage animals. Both the distribution and ratio of monosomes and polysomes were similar among wild type, *eif-3.G(C130Y)*, *acr-2(gf)* and *eif-3.G(C130Y)*; *acr-2(gf)* animals (*Figure 2C–D*), indicating that *eif-3.G(C130Y)* possesses normal function in the majority of tissues. It is possible that *eif-3.G(C130Y)* suppresses *acr-2(gf)* by simply reducing ACR-2 translation. We tested this by examining a functional GFP-tagged ACR-2 single-copy insertion transgene (*oxSi39*). We observed both the levels of ACR-2::GFP fluorescence and post-synaptic localization in ACh-MNs were comparable between wild type and *eif-3.G(C130Y)* animals (*Figure 1—figure supplement 3B*). These data support the conclusion that *eif-3.G(C130Y)* preferentially affects EIF-3's function in ACh-MNs.

## The activity of EIF-3.G(C130Y) requires its RRM

The RRM located at the C-terminus of eIF3g has been shown to bind RNA in a non-specific manner (*Hanachi et al., 1999*). To address the role of the RRM in EIF-3.G's function, we generated a transgene expressing EIF-3.G(ΔRRM) (*Figure 3*; *Supplementary file 1*). Expressing EIF-3.G(ΔRRM) under the endogenous promoter *Peif-3.G* in a wild-type background did not alter development or locomotion, and also did not rescue *eif-3.G(0)* developmental arrest, supporting the essentiality of the EIF-3.G RRM. We then generated a transgene expressing EIF-3.G(C130Y) lacking the RRM domain (C130Y ΔRRM) in neurons of the *acr-2(gf)* background (*Figure 3*). In contrast to full-length *eif-3.G (C130Y)*, *eif-3.G(C130Y ΔRRM)* did not alter convulsion behavior of *acr-2(gf)* mutants (*Figure 3B*), indicating that *eif-3.G(C130Y)* function requires its RRM.

Studies on *S. cerevisiae* TIF35/EIF3.G have shown that its RRM promotes scanning of the translation pre-initiation complex through structured 5'UTRs (*Cuchalová et al., 2010*). Specifically, alanine substitution of three residues in the two ribonucleoprotein (RNP) motifs (K194 in RNP2 and L235 and F237 in RNP1) in TIF35 reduced translation of mRNA reporters carrying 5'UTRs with hairpin structures, without altering the biochemical RNA-binding activity of EIF-3.G/TIF35. Equivalent amino acid residues in *C. elegans* EIF-3.G correspond to R185, F225, F227, which are conserved in human (R242, F282, F284) (*Figure 3*; *Figure 1—figure supplement 1A*). To determine whether these residues affect EIF-3.G's function, we expressed *C. elegans eif-3.G* cDNA with the corresponding amino acids mutated to alanine, designated *eif-3.G(RFF/AAA)*, in *acr-2(gf)* animals. We detected partial suppression of convulsion behavior in *acr-2(gf)* animals (*Figure 3*).

It was also reported that a missense mutation (Q258R) in yeast EIF-3.I/TIF34, located in the sixth WD40 repeat, reduced the rate of pre-initiation complex scanning through 5′UTRs (*Cuchalová et al., 2010*). To test if *C. elegans eif-3.I* shares similar activities, we made a mutant EIF-3.I(E252R), equivalent to yeast TIF34 (Q258R) (*Figure 3*). In *acr-2(gf)* animals, overexpressing *eif-3.I(E252R)*, but not wild-type *eif-3.I(+)*, caused suppression of convulsions to a similar degree as that by the *eif-3.G(RFF/AAA)* transgene (*Figure 3*). These analyses suggest that attenuation of *acr-2(gf)*-induced neuronal overexcitation may involve regulation of protein translation through modification of 5′UTR scanning rates during translation initiation.

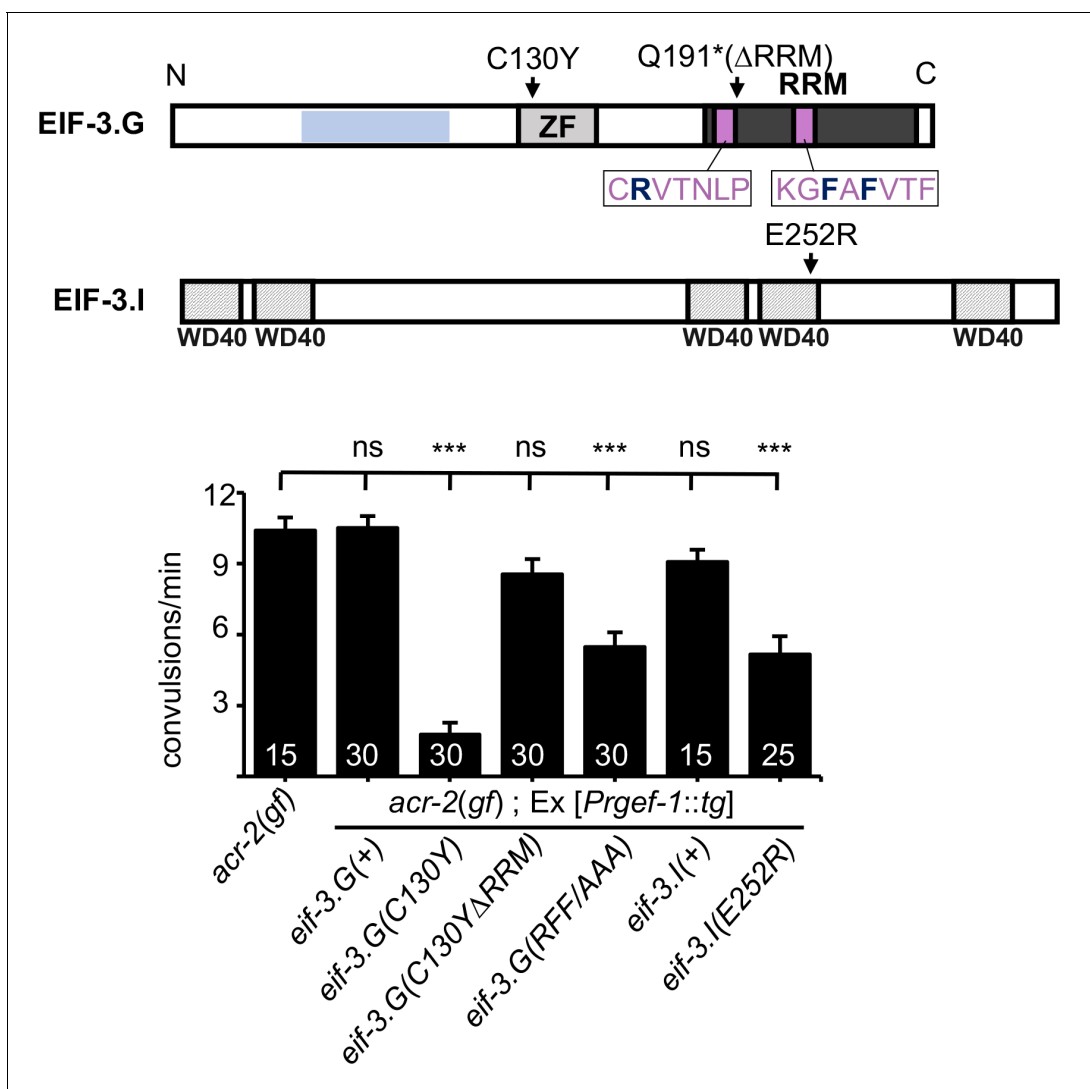

**Figure 3.** *eif-3.G(C130Y)* requires the RNA-binding domain (RRM) to suppress *acr-2(gf)* behaviors. Top illustration of the EIF-3.G protein showing the EIF-3.I binding region (blue), zinc finger (ZF), RRM (dark grey), Q191* mutation in the EIF-3.G(ΔRRM) transgene, RNP motifs (purple), and the RFF residues (bold dark blue) changed to alanine in the *eif-3.G(RFF/AAA)* construct. Below is an illustration of *C. elegans* EIF-3.I pointing to the position of E252R within the fourth WD40 domain. Bottom graph is quantification of convulsion frequency in *acr-2(gf)* animals expressing *eif-3.G* and *eif-3.I* variants in the nervous system (*Prgef-1*). The strains (left to right) are: MT6241, CZ23203/ CZ23204, CZ28152/ CZ28153, CZ23304/ CZ23305, CZ28152/ CZ28153, CZ28057/ CZ28058, CZ28064/ CZ28065. Bars represent mean convulsion frequency ± SEM and sample sizes are indicated within or above bars. (***) p< 0.001, (ns) not significant, by one-way ANOVA with Bonferroni's post-hoc test.

The online version of this article includes the following source data for figure 3:

**Source data 1.** Source data for *Figure 3*.

## Both EIF-3.G(WT) and EIF-3.G(C130Y) associate with mRNA 5′UTRs in the cholinergic motor neurons

EIF-3.G may interact with specific mRNAs in the nervous system to regulate cholinergic activity. Therefore, we next searched for mRNAs that are associated with EIF-3.G(WT) and EIF-3.G(C130Y) in the ACh-MNs using single-end enhanced crosslinking and immunoprecipitation (*Van Nostrand et al., 2017*). We generated single-copy transgenes expressing 3xFLAG-tagged EIF-3.G(WT), EIF-3.G(C130Y), or EIF-3.G(ΔRRM) in the ACh-MNs of *acr-2*(*gf*) animals, with EIF-3.G(ΔRRM) serving to detect indirect crosslinking events. We confirmed that the truncated EIF-3.G(ΔRRM) transgene was expressed, but at reduced levels compared to the EIF-3.G(WT) and EIF-3.G(C130Y) transgenes (*Figure 4—figure supplement 1A*). Following cross-linking and immunoprecipitation using anti-FLAG antibodies, we obtained a comparable amount of immunoprecipitated GFP::EIF-3.G proteins and obtained more reads from seCLIP on animals expressing each GFP::EIF-3.G transgene than on control animals lacking any transgene (IgG(-); see *Supplementary file 4*). There was a strong correlation between read clusters detected among sets of two biological replicates (*Figure 4—figure supplement 1B*). We defined EIF-3.G-RNA crosslink sites as clusters of at least 20 high-quality reads with at least 1.5 fold change enrichment over the input control (see Materials and methods and *Supplementary file 5*). We further defined specific footprints of EIF-3.G(WT) and EIF-3.G(C130Y) by subtracting clusters detected with EIF-3.G(ΔRRM) (*Supplementary file 6*, also see Materials and methods). The EIF-3.G-specific footprints were primarily located within or near the 5′UTRs of protein-coding genes (5′UTR proximal) (*Figure 4A–B*). In total, we detected 231 5′UTR proximal footprints of EIF-3.G(WT) or EIF-3.G(C130Y), which mapped to 225 different genes (*Figure 4C*). The number of reads comprising EIF-3.G(WT) or EIF-3.G(C130Y) footprints was similar (e.g. *egl-30*; *Figure 4B*) for most of these genes. While some footprints were differentially detected between EIF-3.G(WT) and EIF-3.G(C130Y), this was almost invariably due to small differences in seCLIP signal intensity (read cluster size) between samples close to the 20 reads threshold (*Figure 4C*), and we therefore did not further pursue its significance.

In line with a recent report that the human eIF3 complex remains attached to 80S ribosomes in early elongation (*Wagner et al., 2020*), we observed the bulk of read clusters comprising EIF-3.G(WT) and EIF-3.G(C130Y) footprints mapping between (-)150 to (+)200 nucleotides of the start codon (*Figure 4D*). In contrast, the majority of signals comprising 3′UTR footprints of EIF-3.G(WT) and EIF-3.G(C130Y) were dispersed along the first 200 nucleotides downstream of the stop codon (*Figure 4D*). Overall, the footprint map shows that both EIF-3.G(WT) and EIF-3.G(C130Y) predominantly bind to similar locations within or near the 5′UTRs of 225 genes in the ACh-MNs, hereafter named EIF-3.G targets. Taken together with our finding that *eif-3.G*(*C130Y*) requires its RRM to suppress *acr-2*(*gf*), the seCLIP analysis suggests that the C130Y mutation does not dramatically alter the ability of EIF-3.G to associate with these mRNAs in the ACh-MNs.

## EIF-3.G preferentially interacts with long and GC-rich 5′UTR sequences

5′UTR sequences are widely involved in gene-specific regulation of translation (*Pelletier and Sonenberg, 1985*; *Leppek et al., 2018*). We next assessed whether the selective role of EIF-3.G in protein translation might correlate with specific sequence features in the mRNA targets expressed in ACh-MNs by examining the length and GC-content of their 5′UTRs. In *C. elegans*, about 70% of mRNAs are known to undergo trans-splicing, and 5′UTRs of mRNAs with trans-splice leaders are usually short, with a median length of 29nt. We compared the EIF-3.G target gene list with a database containing a compilation of *C. elegans* trans-splice events from ENCODE analyses (*Allen et al., 2011*). We found that 133 of the 225 (59%) EIF-3.G targets are annotated to undergo trans-splicing, which is comparable to that of transcriptome-wide (*Allen et al., 2011*; *Figure 4—figure supplement 2A*), suggesting that trans-splicing events may not contribute to EIF-3.G's selectivity on mRNA targets. Interestingly, we found that the trans-spliced 5′UTRs of these 133 transcripts are significantly longer (median length = 43nt), compared with all trans-spliced 5′UTRs in the *C. elegans* transcriptome (median length = 29nt; n = 6,674) (*Figure 4—figure supplement 2B*). To assess the GC content for EIF-3.G mRNA targets, we then applied a threshold to the cholinergic neuronal transcriptome of *acr-2*(*gf*) (*McCulloch et al., 2020*) defining a 5′UTR as at least 10 nucleotides upstream of ATG, and also selected the longest 5′UTR isoform per gene to avoid redundant analysis of target genes (see Materials and methods). Using this criterion, we identified a 5′UTR for 4573 different genes in the

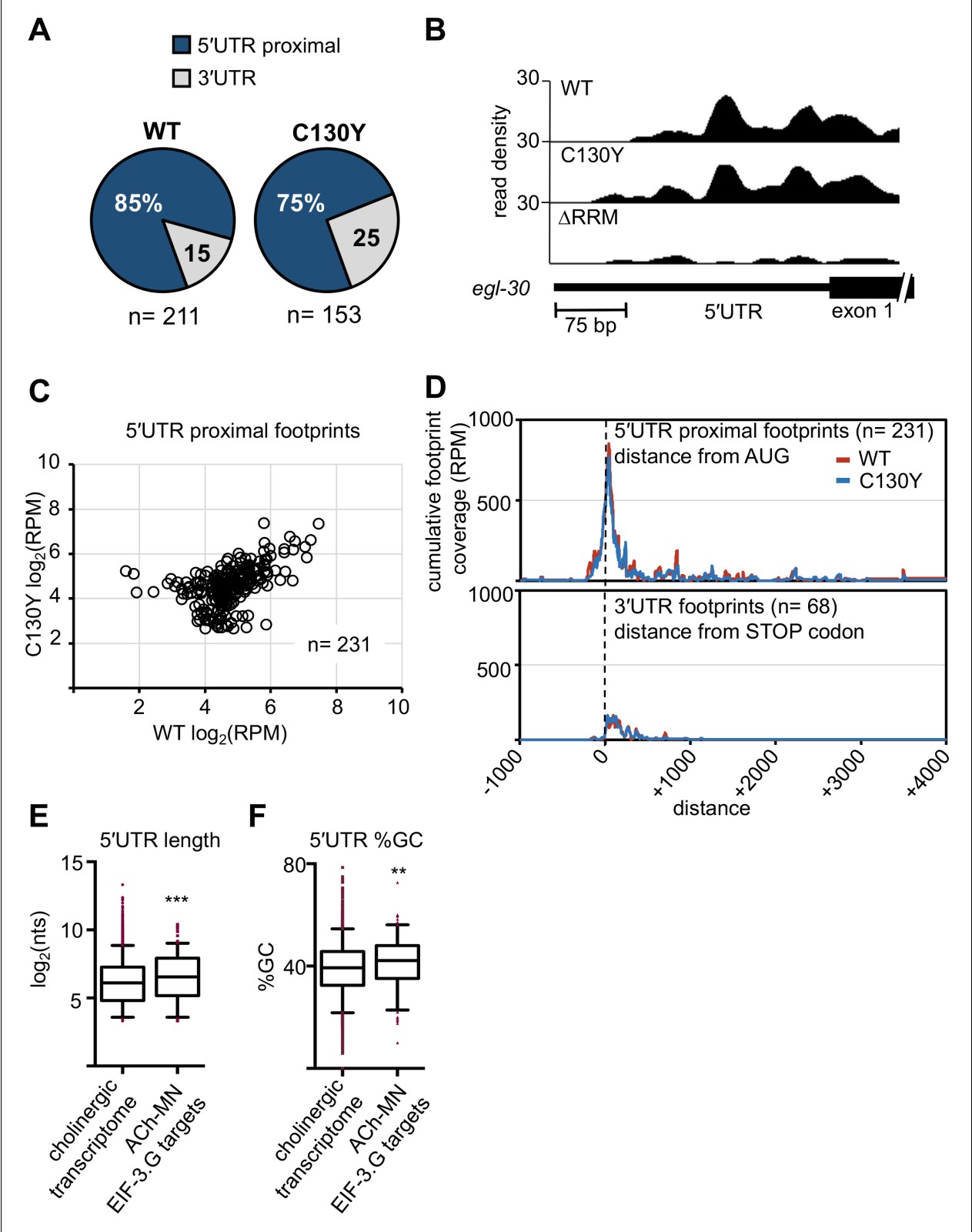

**Figure 4.** Both EIF-3.G(WT) and EIF-3.G(C130Y) associate with mRNA 5′UTRs in the cholinergic motor neurons. (**A**) Pie charts displaying the proportion of EIF-3.G(WT) and EIF-3.G(C130Y) footprints located within each gene feature. (**B**) seCLIP read density track of EIF-3.G(WT) and EIF-3.G(C130Y) footprints on the 5′UTR of *egl-30*, compared to the EIF-3.G(ΔRRM) control. (**C**) Scatter plot comparing the signal intensity, in reads per million (RPM), of all 231 5′UTR proximal footprints detected in EIF-3.G(WT) or EIF-3.G(C130Y). (**D**) Plots show the cumulative coverage of all 5′UTR proximal (top) or

*Figure 4 continued on next page*

*Figure 4 continued*

3'UTR (bottom) footprints of EIF-3.G(WT) or EIF-3.G(C130Y) relative to the start codon (top) or stop codon (bottom) position. Coverage is presented as reads per million (RPM). (E–F) Box plots comparing length and GC-content of all 5'UTR sequences of EIF-3.G target mRNAs with annotations (n = 179) to all 5'UTRs in the *acr-2(gf)* cholinergic neuronal transcriptome (n = 4573). Boxes are 5–95 percentile with outliers aligned in red. Statistics: (***) p< 0.001, (**) p< 0.01 by two-tailed Mann-Whitney test.

The online version of this article includes the following source data and figure supplement(s) for figure 4:

**Source data 1.** Source data for *Figure 4A*.
**Source data 2.** Source data for *Figure 4C*.
**Source data 3.** Source data for *Figure 4D*.
**Source data 4.** Source data for *Figure 4E*.
**Source data 5.** Source data for *Figure 4F*.
**Figure supplement 1.** EIF-3.G transgenes are expressed and produce similar results from replicate seCLIP experiments.
**Figure supplement 1—source data 1.** Source data for *Figure 4—figure supplement 1A*.
**Figure supplement 1—source data 2.** Source data for *Figure 4—figure supplement 1B*.
**Figure supplement 2.** EIF-3.G associates with long and GC-rich 5'UTRs.
**Figure supplement 2—source data 1.** Source data for *Figure 4—figure supplement 2A*.
**Figure supplement 2—source data 2.** Source data for *Figure 4—figure supplement 2B*.
**Figure supplement 2—source data 3.** Source data for *Figure 4—figure supplement 2C*.
**Figure supplement 2—source data 4.** Source data for *Figure 4—figure supplement 2D*.
**Figure supplement 2—source data 5.** Source data for *Figure 4—figure supplement 2E*.

cholinergic transcriptome and for 179 of the 232 EIF-3.G targets in the ACh-MNs. The median 5'UTR among the 179 EIF-3.G target mRNAs was significantly longer (93 nt) and GC-enriched (42%), compared to the cholinergic transcriptome median (69 nt and 39% GC; n = 10,962; *Figure 4E–F*). We further analyzed the distribution of GC sequences in 5'UTRs, and observed non-random positioning such that some genes were relatively GC-rich near the start codon (e.g. *zip-2* and *sec-61*) and others had enrichment closer to the distal 5' end (e.g. *pdf-1* and *kin-10*), suggesting that discrete sequence elements in EIF-3.G associated transcripts may regulate translation (*Figure 4—figure supplement 2C*).

The incidence of long and GC-enriched 5'UTRs among EIF-3.G associated transcripts led us to speculate a major function of EIF-3.G, in addition to its necessity in general translation initiation, is in the selective regulation of translation. To extend our findings beyond *C. elegans*, we asked if the preferential association of EIF-3.G with these complex 5'UTRs could be conserved in mammals. We analyzed the published eIF3g PAR-CLIP sequencing data from HEK293 cells (*Lee et al., 2015*) by comparing the 5'UTR lengths of human eIF3g target genes to all genes with 5'UTRs annotated in the hg38 genome. We found that human transcripts associated with eIF3g contained significantly longer and GC-enriched 5'UTRs than average (*Figure 4—figure supplement 2D–E*). This analysis lends support for a conserved, specialized role of eIF3g in the translation of transcripts harboring complex 5'UTRs.

## EIF-3.G target mRNAs encode proteins that exhibit activity-dependent expression

To address whether EIF-3.G target mRNAs may preferentially affect specific biological processes, we performed Gene Ontology and KEGG pathway analysis. Significant GO term (*Ashburner et al., 2000*) enrichment was identified in neuropeptide signaling genes (GO:0050793; 15 genes), which are known to affect *acr-2*(*gf*) behavior (*Stawicki et al., 2013*; *McCulloch et al., 2020*), and in stress response genes (GO: 0006950; 28 genes), which could modulate neuronal homeostasis or function under circuit activity changes (*Figure 5A*). We also found many EIF-3.G target genes involved in protein translation and protein metabolism processes (GO:0019538; 29 genes; *Figure 5A*). Additional enrichment was associated with metabolic components, kinase signaling, and calcium and synaptic signaling pathways (*Figure 5A*). Calcium and synaptic signaling genes included the CAMKII *unc-43*, and the G-proteins *egl-30* and *goa-1*, which are all known to regulate ACh-MN synaptic activity (*Miller et al., 1999*; *Richmond, 2005*; *Treinin and Jin, 2020*).

To determine if expression of EIF-3.G target mRNAs is regulated in an activity-dependent manner, we next incorporated differential transcript expression data between wild type and *acr-2(gf)*

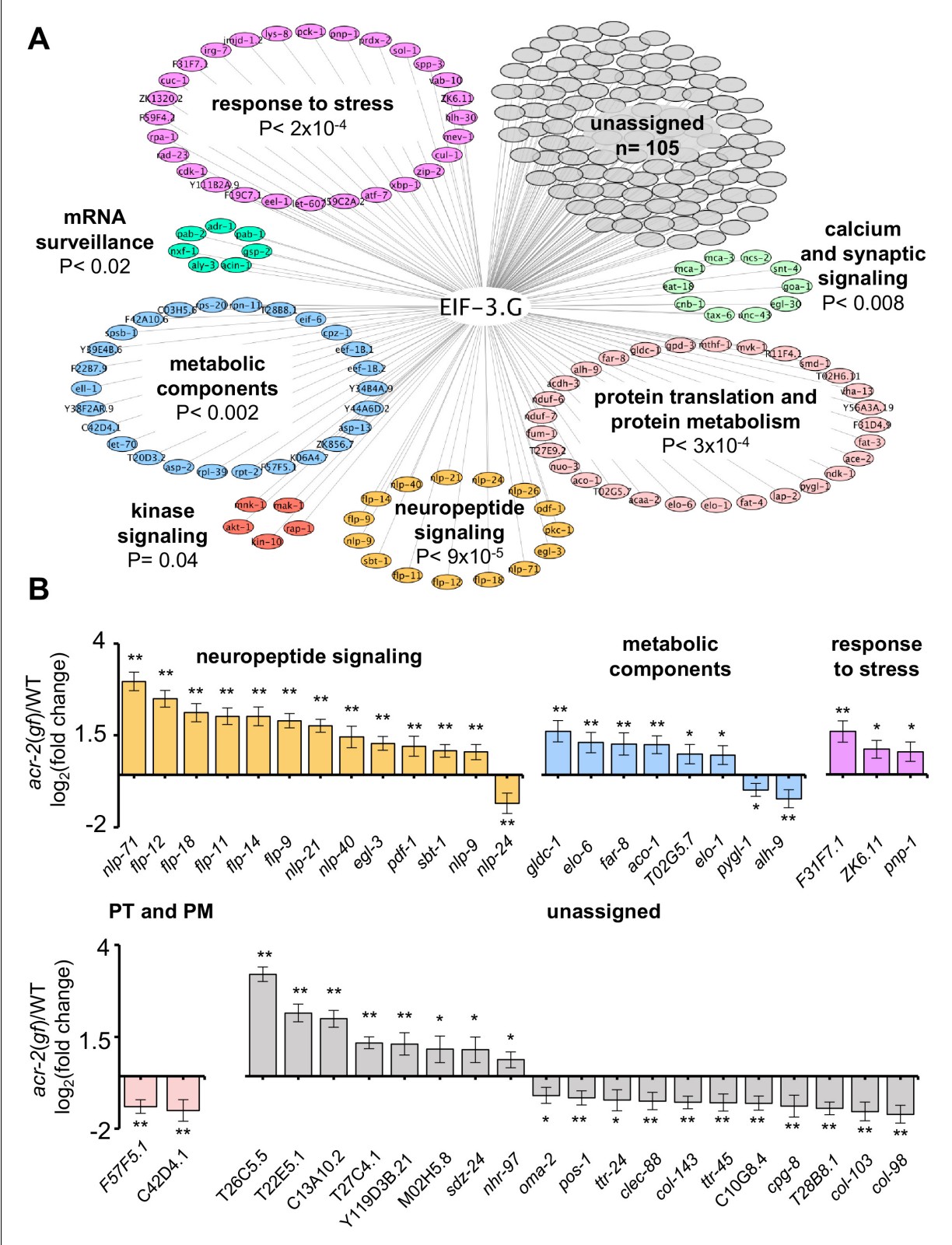

**Figure 5.** Gene network analyses of EIF-3.G target mRNAs show enrichment in activity-dependent expression. (**A**) Cytoscape network of EIF-3.G target genes with enriched GO terms (neuropeptide signaling, response to stress, and protein translation and protein metabolism) or KEGG pathways (calcium and synaptic signaling, metabolic components, MAPK-signaling, and mRNA surveillance). Enrichment p-values are derived from statistical analysis of our EIF-3.G targets (n = 225) in the PANTHER database (***Mi et al., 2019***). (**B**) EIF-3.G target genes exhibiting significant transcript level
*Figure 5 continued on next page*

Figure 5 continued

changes in *acr-2*(*gf*) versus wild-type animals as determined from transcriptome sequencing of cholinergic neurons by McCulloch et al. PT and PM refers to protein translation and protein metabolism. Differential expression was assessed using DeSeq2 (*Love et al., 2014*) with significance thresholds of (*) p<0.05 and (**) p<0.01.

The online version of this article includes the following source data for figure 5:

**Source data 1.** Source data for *Figure 5B*.

from a cholinergic neuron transcriptome dataset (*McCulloch et al., 2020*). We found that 83% of EIF-3.G target mRNAs in the ACh-MNs are present in the cholinergic neuron transcriptome. Among the 45 genes exhibiting significant expression changes dependent on *acr-2*(*gf*) (*Figure 5B*), nearly all neuropeptide signaling transcripts (12 of 15) as well as three stress response genes were upregulated in *acr-2*(*gf*) (*Figure 5B*). Genes encoding metabolic components were variably upregulated (e.g. Glycine decarboxylate/*gldc-1*, aconitase/*aco-1*) and downregulated (e.g. glycogen phosphorylase/*pygl-1*, aldehyde dehydrogenase/*alh-9*) (*Figure 5B*). These data support the idea that wild type EIF-3.G imparts translational control to activity-dependent expression changes and that EIF-3.G (C130Y) may exert specific regulation to alter their protein expression in ACh-MNs of *acr-2*(*gf*).

## EIF-3.G modulates translation of HLH-30 and NCS-2 in hyperactive ACh-MNs

To experimentally validate that EIF-3.G regulates protein expression from its target mRNAs in the ACh-MNs, we next surveyed a number of candidate genes, chosen mainly based on the availability of transgenic reporters that contain endogenous 5′UTRs (*Supplementary file 1*). We identified two genes (*hlh-30* and *ncs-2*) whose expression in ACh-MNs of *acr-2*(*gf*) animals shows dependency on EIF-3.G. *hlh-30* produces multiple mRNA isoforms (*Figure 6A*), which encode the *C. elegans* ortholog of the TFEB stress response transcription factor with broad neuroprotective roles (*Decressac et al., 2013*; *Polito et al., 2014*; *Lin et al., 2018*). We observed strong seCLIP signals corresponding to EIF-3.G(WT) and EIF-3.G(C130Y) footprints in the 5′UTR of long isoform d, but not in isoform a (*Figure 6B*). The *hlh-30d* mRNA isoform has a 5′UTR of 190nt with 43% GC. Using computational RNA structure prediction (RNAfold), we found that the long *hlh-30d* 5′UTR forms strong stem-loop structures (ΔG = −40.78 kcal/mol) that could affect HLH-30 translation. We examined expression of an HLH-30::EGFP fosmid reporter *wgIs433*, which encompasses the entire *hlh-30* genomic region with *cis*-regulatory elements for all mRNA isoforms (*Sarov et al., 2006*; *Figure 6C*). HLH-30::GFP was observed throughout the nervous system and primarily localized to cytoplasm in all genetic backgrounds tested. We observed significantly enhanced HLH-30::GFP signals in the ACh-MNs of *acr-2*(*gf*) animals, compared to those in wild type (*Figure 6C*). While *eif-3.G*(*C130Y*) did not alter HLH-30::GFP, it reduced fluorescence intensity in *acr-2*(*gf*) to wild type levels (*Figure 6C*). As *hlh-30* transcripts were detected at similar levels in ACh-MNs of wild type and *acr-2*(*gf*) animals (*McCulloch et al., 2020*), the enhanced HLH-30::GFP signal in *acr-2*(*gf*) likely reflects elevated translation upon neuronal activity changes, which is augmented by EIF-3.G. To strengthen this idea, we introduced an *unc-13* null allele, which blocks presynaptic release (*Richmond et al., 1999*) to the above analyzed compound genetic mutants. We found that *unc-13*(*0*) abolished the enhanced HLH-30::GFP expression caused by *acr-2*(*gf*) (*Figure 6C*). Additionally, we tested a transgenic HLH-30a::GFP reporter expressing *hlh-30a* cDNA driven by the 2 kb sequence upstream of that isoform (*Figure 6—figure supplement 1*).

We found that HLH-30a::GFP intensity was comparable between *acr-2*(*gf*) and *eif-3.G*(*C130Y*); *acr-2*(*gf*) (*Figure 6D*). These data strengthen the conclusion that enhanced HLH-30 translation in *acr-2*(*gf*) partly involves the complex 5′UTR of *hlh-30d*.

The Neuronal Calcium Sensor protein encoded by *ncs-2* promotes calcium-dependent signaling in ACh-MNs (*Zhou et al., 2017*). We identified strong and specific association of EIF-3.G(WT) and EIF-3.G(C130Y) overlapping the 5′UTR of *ncs-2* (*Figure 7A*). To evaluate NCS-2 expression, we examined a single-copy translational reporter (*juSi260*) expressing NCS-2::GFP under its endogenous promoter (*Zhou et al., 2017*; *Figure 7B*). NCS-2::GFP localized primarily to the neuronal processes in ventral nerve cord, because of the N-terminal myristoylation motif. Quantification of NCS-2::GFP showed that the fluorescence intensity in *eif-3.G*(*C130Y*); *acr-2*(*gf*) double mutants was significantly

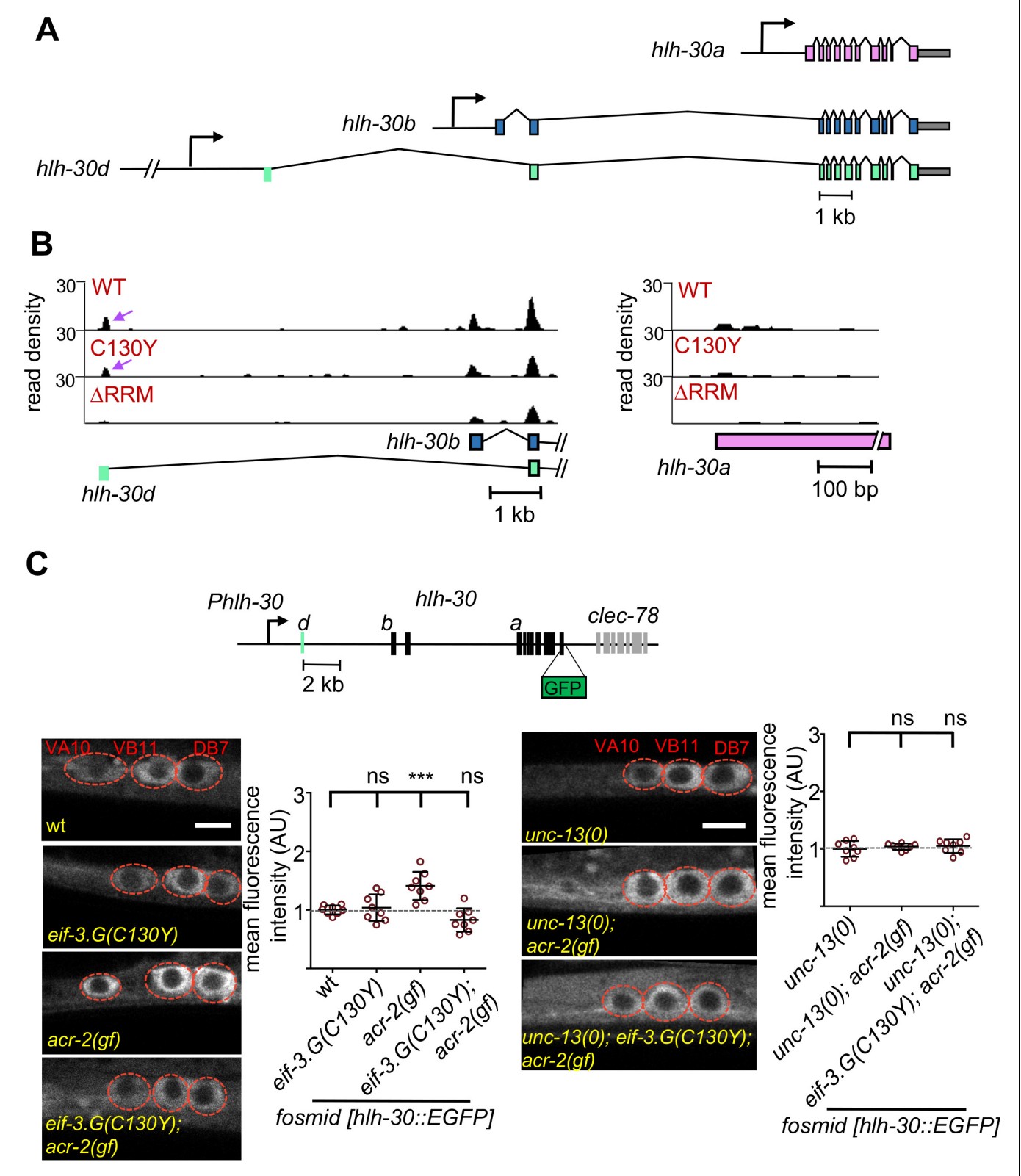

**Figure 6.** EIF-3.G(C130Y) impairs HLH-30 expression in ACh-MNs of *acr-2*(*gf*) animals. (**A**) Gene models of *hlh-30* isoforms *a* (pink), *b* (blue), and *d* (green), with presumptive promoters for each isoform depicted as right-pointing arrows and the 5′UTR of *isoform d* in green to the right of its promoter. (**B**) seCLIP read density tracks of footprints on the 5′ end of *hlh-30 isoform b* and *d* (left) and the 5′ end of *hlh-30 isoform a* (right) in each indicated EIF-3.G dataset. Purple arrows show footprints on the 5′UTR of *hlh-30 isoform d*. (**C**) Top: Illustration of the *wgIs433* fosmid locus with *hlh-30*

*Figure 6 continued on next page*

*Figure 6 continued*

coding exons in black and 5′UTR of *isoform d* in green to the right of the promoter. Bottom: Representative single-plane confocal images of the fosmid translational reporter *wgIs433*[*hlh-30*::EGFP::3xFLAG] in ACh-MNs in animals of indicated genotypes. Quantification of GFP intensity is shown on the right (n = eight for each genotype). Animals are oriented with anterior to the left. Scale bar = 4 μm. Red dashes indicate labeled ACh-MN soma. Each data point is the average fluorescence intensity quantified from the three ACh-MN soma per animal and normalized to the mean intensity obtained from *wgIs433* in the wild type background. Statistics: (***) p< 0.001, (ns) not significant, one-way Anova with Bonferroni's post hoc test.

The online version of this article includes the following source data and figure supplement(s) for figure 6:

**Source data 1.** Source data for *Figure 6C*.
**Figure supplement 1.** EIF-3.G (C130Y) has no effect on translation of *hlh-30.a* in the ACh-MNs.
**Figure supplement 1—source data 1.** Source data for *Figure 6—figure supplement 1*.

reduced, compared to those in wild type, *eif-3.G(C130Y)*, and *acr-2(gf)* (*Figure 7B*). *ncs-2* mRNA is SL1 trans-spliced, and the mature 5′UTR has 37 nt that is especially abundant in GC nucleotides (47% GC) (*Figure 7B*). Moreover, the *ncs-2* 5′UTR sequence is highly conserved with other nematode species (*Figure 7—figure supplement 1A*). By RNAfold prediction, we found this sequence could form a strong stem-loop structure (ΔG = −5.10 kcal/mol). To test if NCS-2::GFP expression was regulated specifically through its 5′UTR, we replaced it with the 5′UTR of *eif-3.G*, which is comparatively reduced in GC-content (37% GC) and with much less folding stability (Δ = −1.95 kcal/mol) (*Figure 7C*). The *eif-3.G* 5′UTR is also less conserved across nematodes compared to that of *ncs-2* (*Figure 7—figure supplement 1A*). We found that the NCS-2::GFP reporter with the 5′UTR of *eif-3. G* was expressed at similar levels in all genetic backgrounds (*Figure 7C*).

To further determine the effects of the *ncs-2* 5′UTR in protein translation with neuronal type resolution, we generated a reporter in which the GFP coding sequence was fused in-frame after the first four amino acids of NCS-2, which retains the *ncs-2* 5′UTR but disrupts the myristoylation motif, thereby enabling visualization of NCS-2 in ACh-MNs (*Figure 7—figure supplement 1B*). Quantification of GFP fluorescence in the cell bodies of VA10, VB11, and DB7 ACh-MN showed significantly reduced expression in *eif-3.G(C130Y)*; *acr-2(gf)* animals (*Figure 7—figure supplement 1B*). In contrast, a similar reporter but with the 5′UTR of *eif-3.G* displayed similar GFP levels in all genetic backgrounds (*Figure 7—figure supplement 1C*). Therefore, we conclude that *eif-3.G* regulates NCS-2 expression in the ACh-MNs through a mechanism involving its 5′UTR sequence.

## Discussion

The eIF3 complex has been extensively studied for its essential roles in general translation initiation (*Cate, 2017*; *Valášek et al., 2017*). However, recent work gives support to the idea that eIF3 is also key to many of the specialized translational control mechanisms needed for tissue plasticity in vivo (*Lee et al., 2015*; *Shah et al., 2016*; *Rode et al., 2018*; *Lamper et al., 2020*). Our work expands the landscape of eIF3's regulatory functions, revealing an in vivo role of the eIF3g subunit in stimulating the translation of proteins that mediate neuronal activity changes.

### EIF-3.G ensures the efficient translation of mRNAs with GC-rich 5′UTRs

Our study is the first application of seCLIP-seq to map transcriptome-wide protein binding sites in a specific neuronal subtype (ACh-MNs) in *C. elegans*. With stringent thresholding, we identified 225 genes with strong EIF-3.G occupancy at mRNA 5′ ends. We find that EIF-3.G generally associates with mRNAs harboring long and GC-rich 5′UTRs, implying its RNA-binding function is selective for stimulating translation initiation on 5′ leaders prone to secondary structure or other forms of translation regulation. Our data provide in vivo support to the finding that yeast eIF3g/TIF35 promotes scanning through 5′UTRs with stem-loop structures (*Cuchalová et al., 2010*). The RRM of yeast eIF3g/TIF35 also promotes re-initiation of 40S ribosomes upon terminating at uORF stop codons on GCN4, thereby allowing efficient induction of genes whose translation is regulated by uORFs (*Cuchalová et al., 2010*). We did not observe uORFs in the 5′UTRs of *ncs-2* or *hlh-30*, suggesting that at least for these mRNAs, *eif-3.G(C130Y)* involves reduced scanning through secondary structures or other yet undefined regulatory sequence elements.

It is worth noting that we also found EIF-3.G footprints in 3′UTRs, which could reflect molecular crosstalk between translation initiation and 3′UTR factors, given their proximity in the closed loop

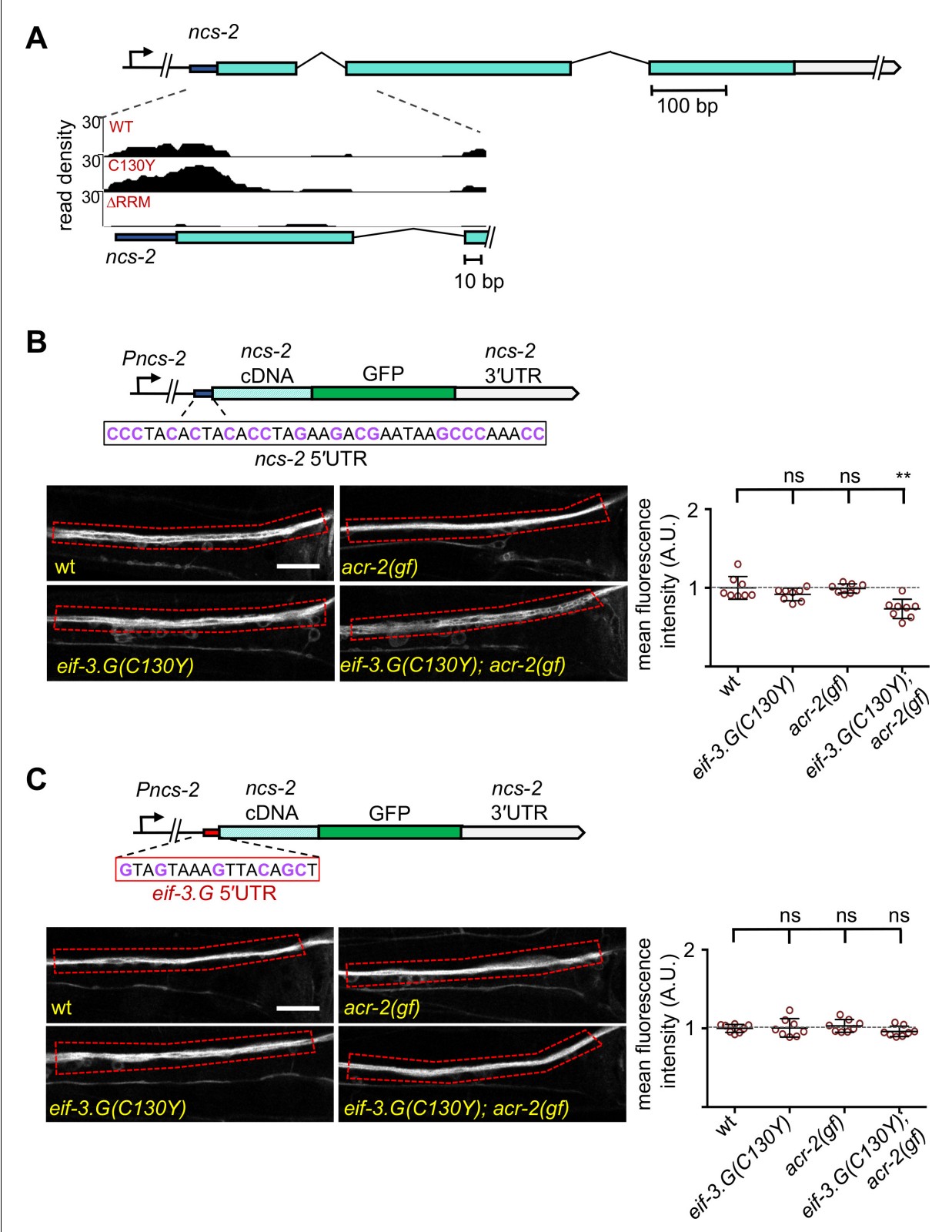

**Figure 7.** Regulation of NCS-2 expression by EIF-3.G depends on its GC-rich 5′UTR. (**A**) Illustration of the *ncs-2* genomic region. Dark blue represents 5′UTR, green boxes are coding exons, and gray is the 3′UTR. The inset below shows the read density track of seCLIP footprints on the 5′ region of *ncs-2* mRNA. (**B**) Top: Schematic of the NCS-2(cDNA)::GFP translation reporter, including its 5′UTR (dark blue), driven by the 4 kb promoter *Pncs-2*. The 5′UTR sequences are GC rich (purple). *Bottom:* Representative single-plane confocal images of NCS-2::GFP in ventral nerve chord processes in young adult

*Figure 7 continued on next page*

*Figure 7 continued*

animals of the indicated genotypes. GFP intensity quantification is shown to the right. (**C**) Top: The *ncs-2*(5′UTR mutant)::GFP translational reporter has the 5′UTR of *eif-3.G* (red boxed sequence) replacing the *ncs-2* 5′UTR, driven by *Pncs-2*. Bottom: Representative single-plane confocal images of ventral nerve chord processes expressing the NCS-2(5′UTR mutant)::GFP translation reporter in young adult animals of the indicated genotypes. GFP intensity quantification is shown to the right. For (**B**) and (**C**), data points are normalized to the average fluorescence intensity of the respective translation reporter in the wild-type background. ROIs used for fluorescence quantification are boxed. Scale bar = 15 μm. Statistics: (**) P< 0.01, (ns) not significant by one-way Anova with Bonferroni's post hoc test.

The online version of this article includes the following source data and figure supplement(s) for figure 7:

**Source data 1.** Source data for *Figure 7B*.
**Source data 2.** Source data for *Figure 7C*.
**Figure supplement 1.** EIF-3.G(C130Y) reduces NCS-2 expression in the ACh-MNs of *acr-2*(*gf*) animals dependent on its conserved 5′UTR.
**Figure supplement 1—source data 1.** Quantification of relative fluorescence intensity in the indicated strains.
**Figure supplement 1—source data 2.** Quantification of relative fluorescence intensity in the indicated strains.

translation model (*Imataka et al., 1998*; *Wells et al., 1998*). EIF-3.G might anchor the closed-loop mRNA form that stimulates multiple rounds of translation, as was shown to be the case with eIF3h (*Choe et al., 2018*). It is also possible that EIF-3.G cooperates with 3′UTR interacting factors that regulate gene expression, as several *C. elegans* translation initiation factors co-immunoprecipitated with the miRISC complex (*Zhang et al., 2007*) and accumulating evidence supports interplay between various translation factors and RISC proteins that mediate translational repression by micro-RNAs (*Ricci et al., 2013*; *Fukaya et al., 2014*; *Gu et al., 2014*). Thus, further analysis is needed to examine the biological meaning of EIF-3.G association with 3′UTRs.

## The EIF-3.G zinc finger conveys a selective function to translation initiation

The function of the zinc finger of eIF3g remains undefined. Through analysis of EIF-3.G(C130Y), our data provides in vivo insights that the zinc finger contributes to translation efficiency of mRNAs harboring complex 5′UTRs. We establish that EIF-3.G(C130Y) behaves as a genetic gain-of-function mutation without disrupting EIF-3 assembly or otherwise impairing general translation, measured by both polysome levels and the health of cells, tissues, and animals. Additionally, mutating a different cysteine within the zinc finger (C127Y) causes equivalent effects, further strengthening the important role of the entire zinc finger. The effect of EIF-3.G(C130Y) on *acr-2*(*gf*) behaviors depends on the RRM, suggesting that association with mRNA after assembly of the pre-initiation complex is required for EIF-3.G(C130Y) function. While we did not observe significant mis-positioning of EIF-3.G-mRNA interactions by EIF-3.G(C130Y), we acknowledge that seCLIP may not have the resolution required to reveal subtle differences in crosslinking sites caused by the C130Y alteration. Together, our data is consistent with a model where EIF-3.G(C130Y) imposes a translational stall after EIF-3 complex assembly and mRNA recruitment. In this view, we speculate that the zinc finger of EIF-3.G mediates interactions with other proteins, such as the ribosome, that critically regulate translation events after mRNA binding. In support of this model, yeast eIF3g/TIF35 was found to directly bind to small ribosomal protein RPS-3, though the molecular basis for mediating this interaction is not identified (*Cuchalová et al., 2010*). Further studies are required to address the precise molecular mechanism by which the EIF-3.G zinc finger imparts regulatory control over translation initiation.

## EIF-3.G targets the translation of mRNAs that modulate neuronal function

Our study was driven by the genetic evidence that *eif-3.G*(*C130Y*) ameliorates convulsion behavior caused by the hyperactive ion channel ACR-2(GF). We show that EIF-3.G(C130Y) retains essential EIF-3.G function, yet it alters protein translation on select mRNAs in hyperactive ACh-MNs, as evidenced by its effects on NCS-2 and HLH-30 expression. We previously reported that complete loss-of-function of *ncs-2* strongly suppresses *acr-2*(*gf*) behaviors to a similar degree as *eif-3.G*(*C130Y*) (*Zhou et al., 2017*). However, 50% reduction of *ncs-2* expression does not cause detectable consequences and complete loss-of-function in *hlh-30* also has no effects in either wild type or *acr-2*(*gf*). Thus, the small reduction of NCS-2 and HLH-30 waged by *eif-3.G*(*C130Y*) is unlikely to account for the full extent of phenotypic suppression of *acr-2*(*gf*). Our seCLIP data also revealed EIF-3.G

interactions with many other genes that differentially impact *acr-2*(*gf*) behavior (e.g. neuropeptide *flp-18*, endopeptidase *egl-3*) and cholinergic activity (e.g. G proteins *goa-1*, *egl-30*). Interestingly, many of the pre-synaptic genes that regulate *acr-2*(*gf*) behavior, such as *unc-13*/Munc13, *unc-17*/VAChT (*Zhou et al., 2013*; *Takayanagi-Kiya et al., 2016*; *McCulloch et al., 2017*), do not have EIF-3.G footprints. Thus, our data is consistent with a model where *eif-3.G*(*C130Y*) ameliorates behaviors of *acr-2*(*gf*) through the cumulative changes of select ACh-MN activity regulators.

## *eif-3.G* function may be specialized for activity-dependent gene expression

The eIF3 complex is widely implicated in brain disorders, and deregulated eIF3g is specifically linked to narcolepsy (*Gomes-Duarte et al., 2018*). However, given the essential role of eIF3 in protein translation in all tissues, investigation of its functions in the nervous system remains limited. Our results reveal that EIF-3.G permits normal activity-dependent protein expression changes, and suggest that dysregulated EIF-3.G might potentiate aberrant neuronal behavior in disorders such as epilepsy by altering the neuronal protein landscape. It is worth noting that pore-lining mutations in human nicotinic receptors that occur at similar positions as *acr-2*(*gf*) are causally linked to epilepsy (*Xu et al., 2011*). We speculate that EIF-3.G may be a potential target for intervention of disorders involving abnormal neurological activity.

In summary, our findings echo the general notion that fine-tuning the activity of essential cellular machinery, such as ribosomes and translation complexes holds the key to balance cellular proteome under dynamic environmental challenges or disease conditions. Emerging studies from cell lines show that stress conditions can induce post-translational modification of eIF3 subunits (*Lamper et al., 2020*) or cap-independent interactions with mRNAs to modify proteomes (*Meyer et al., 2015*). Through characterization of the G subunit of eIF3, we reveal the first mechanistic insights into how the eIF3 complex regulates neuronal activity. It is likely that individual eIF3 subunits could each possess unique functions relevant in certain contexts, altogether providing the eIF3 complex with extensive utility to remodel the proteome in response to changing cellular environments.

# Materials and methods

**Key resources table**

| Reagent type (species) or resource | Designation | Source or reference | Identifiers | Additional information |
|---|---|---|---|---|
| Antibody | anti-FLAG (Rabbit) | Sigma-Aldrich | Cat# F7425, RRID:AB_439687 | WB (1:2000) |
| Antibody | anti-Actin clone C4 (Mouse monoclonal) | MP Biomedicals | Cat# 08691002, RRID:AB_2335304 | WB (1:2000) |
| Antibody | Anti-FLAG M2 Magnetic Beads | Sigma-Aldrich | Cat# M8823, RRID:AB_2637089 | IP |
| Recombinant protein reagent | Cas9-NLS (purified protein) | UC Berkely QB3 | | |
| Genetic reagent (*C. elegans*) | + | CGC | RRID:CGC_N2 | |
| Genetic reagent (*C. elegans*) | *acr-2*(*n2420*) X | *Jospin et al., 2009* | MT6241 | |
| Genetic reagent (*C. elegans*) | *eif-3.G*(*ju807*) II | This work | CZ22197 | *Figure 1F* |
| Genetic reagent (*C. elegans*) | *eif-3.G*(*ju1840*) II | This work | CZ28494 | *Figure 1C* |
| Genetic reagent (*C. elegans*) | *eif-3.G*(*ju807*) II; *acr-2*(*n2420*) X | This work | CZ21759 | *Figure 1C* |
| Genetic reagent (*C. elegans*) | *eif-3.G*(*ju1840*) II; *acr-2*(*n2420*) X | This work | CZ28495 | *Figure 1C* |

*Continued on next page*

*Continued*

| Reagent type (species) or resource | Designation | Source or reference | Identifiers | Additional information |
|---|---|---|---|---|
| Genetic reagent (C. elegans) | eif-3.G(ju1327) / mnC1 II | This work | CZ22974 | *Figure 1F* |
| Genetic reagent (C. elegans) | acr-2(n2420) X; juEx7015 | This work | CZ22976 | *Figure 1C* |
| Genetic reagent (C. elegans) | acr-2(n2420) X; juEx7016 | This work | CZ22977 | *Figure 1C* |
| Genetic reagent (C. elegans) | eif-3.G(ju807) I; acr-2(n2420) X; juEx7045 | This work | CZ23125 | *Figure 1E* |
| Genetic reagent (C. elegans) | eif-3.G(ju807) II; acr-2(n2420) X; juEx7046 | This work | CZ23126 | *Figure 1E* |
| Genetic reagent (C. elegans) | eif-3.G(ju807) II; acr-2(n2420) X; juEx7019 | This work | CZ22980 | *Figure 1E* |
| Genetic reagent (C. elegans) | eif-3.G(ju807) II; acr-2(n2420) X; juEx7020 | This work | CZ22981 | *Figure 1E* |
| Genetic reagent (C. elegans) | eif-3.G(ju807) II; acr-2(n2420) X; juEx7439 | This work | CZ23791 | *Figure 1E* |
| Genetic reagent (C. elegans) | eif-3.G(ju807) II; acr-2(n2420) X; juEx7440 | This work | CZ23880 | *Figure 1E* |
| Genetic reagent (C. elegans) | eif-3.G(ju807) II; acr-2(n2420) X; juEx7021 | This work | CZ22982 | *Figure 1E* |
| Genetic reagent (C. elegans) | eif-3.G(ju807) II; acr-2(n2420) X; juEx7022 | This work | CZ22983 | *Figure 1E* |
| Genetic reagent (C. elegans) | eif-3.G(ju807) II; acr-2(n2420) X; juEx8062 | This work | CZ27881 | *Figure 1E* |
| Genetic reagent (C. elegans) | eif-3.G(ju807) II; acr-2(n2420) X; juEx8063 | This work | CZ27882 | *Figure 1E* |
| Genetic reagent (C. elegans) | eif-3.G(ju1327) /mnC1 II; acr-2(n2420) X | This work | CZ23310 | *Figure 1F* |
| Genetic reagent (C. elegans) | eif-3.G(ju807) / eif-3.G(ju1327) II | This work | CZ25714 | *Figure 1F* |
| Genetic reagent (C. elegans) | eif-3.G(ju807) / eif-3.G(ju1327) II; acr-2(n2420) X | This work | CZ26828 | *Figure 1F* |
| Genetic reagent (C. elegans) | juSi320 IV | This work | CZ24063 | *Figure 2B*; *Figure 1—figure supplement 1B* |
| Genetic reagent (C. elegans) | eif-3.G(ju1327) /mnC1 II; juSi320 IV | This work | CZ24079 | *Figure 2A* |
| Genetic reagent (C. elegans) | juSi320 IV; acr-2(n2420) X | This work | CZ24729 | *Figure 2A* |
| Genetic reagent (C. elegans) | eif-3.G(ju807) II; juSi320 IV; acr-2(n2420) X | This work | CZ28107 | *Figure 2A* |
| Genetic reagent (C. elegans) | juSi331 IV | This work | CZ24651 | *Figure 2B*; *Figure 1—figure supplement 1B* |
| Genetic reagent (C. elegans) | juSi331 IV; acr-2(n2420) X | This work | CZ24652 | *Figure 2A* |
| Genetic reagent (C. elegans) | eif-3.G(ju1327) / mnC1 II; juSi331 IV; acr-2(n2420) X | This work | CZ28497 | *Figure 2A* |
| Genetic reagent (C. elegans) | juIs14 IV | *Wang et al., 2017* | CZ631 | |
| Genetic reagent (C. elegans) | eif-3.G(ju807) II; juIs14 IV | This work | CZ24161 | *Figure 1—figure supplement 2* |
| Genetic reagent (C. elegans) | juIs14 IV; acr-2(n2420) X | *McCulloch et al., 2020* | CZ5808 | |

*Continued on next page*

*Continued*

| Reagent type (species) or resource | Designation | Source or reference | Identifiers | Additional information |
|---|---|---|---|---|
| Genetic reagent (*C. elegans*) | *eif-3.G(ju807)* II; *juIs14* IV; *acr-2(n2420)* X | This work | CZ8905 | ***Figure 1—figure supplement 2A*** |
| Genetic reagent (*C. elegans*) | *nuIs94* | ***Hallam et al., 2000*** | KP2229 | |
| Genetic reagent (*C. elegans*) | *eif-3.G(ju807)* II; *nuIs94* | This work | CZ24021 | ***Figure 1—figure supplement 2B*** |
| Genetic reagent (*C. elegans*) | *acr-2(n2420)* X; *nuIs94* | This work | CZ5815 | ***Figure 1—figure supplement 2B*** |
| Genetic reagent (*C. elegans*) | *eif-3.G(ju807)* II; *acr-2(n2420)*X; *nuIs94* | This work | CZ24021 | ***Figure 1—figure supplement 2B*** |
| Genetic reagent (*C. elegans*) | *eif-3.E(ok2607)* I / hT2 I,III; *acr-2(n2420)* X | This work | CZ27434 | ***Figure 1—figure supplement 3A*** |
| Genetic reagent (*C. elegans*) | *eif-3.E(ok2607)* I / hT2 I, III; *eif-3.G(ju807)* II; *acr-2(n2420)* X | This work | CZ27433 | ***Figure 1—figure supplement 3A*** |
| Genetic reagent (*C. elegans*) | *eif-3.H(ok1353)* I / hT2 I, III; *acr-2(n2420)* X | This work | CZ27435 | ***Figure 1—figure supplement 3A*** |
| Genetic reagent (*C. elegans*) | *eif-3.H(ok1353)* I / hT2 I, III; *eif-3.G(ju807)* II; *acr-2(n2420)* X | This work | CZ27436 | ***Figure 1—figure supplement 3A*** |
| Genetic reagent (*C. elegans*) | *oxSi39* IV | ***Qi et al., 2013*** | CZ12338 | |
| Genetic reagent (*C. elegans*) | *eif-3.G(ju807)* II; *oxSi39* IV | This work | CZ23854 | ***Figure 1—figure supplement 3B*** |
| Genetic reagent (*C. elegans*) | *acr-2(n2420)* X; *juEx7056* | This work | CZ23203 | ***Figure 3*** |
| Genetic reagent (*C. elegans*) | *acr-2(n2420)* X; *juEx7057* | This work | CZ23204 | ***Figure 3*** |
| Genetic reagent (*C. elegans*) | *acr-2(n2420)* X; *juEx8100* | This work | CZ28152 | ***Figure 3*** |
| Genetic reagent (*C. elegans*) | *acr-2(n2420)* X; *juEx8101* | This work | CZ28153 | ***Figure 3*** |
| Genetic reagent (*C. elegans*) | *juEx7113* | This work | CZ26777 | ***Figure 3*** |
| Genetic reagent (*C. elegans*) | *acr-2(n2420)* X; *juEx7114* | This work | CZ23304 | ***Figure 3*** |
| Genetic reagent (*C. elegans*) | *acr-2(n2420)* X; *juEx7115* | This work | CZ23305 | ***Figure 3*** |
| Genetic reagent (*C. elegans*) | *acr-2(n2420)* X; *juEx8095* | This work | CZ28066 | ***Figure 3*** |
| Genetic reagent (*C. elegans*) | *acr-2(n2420)* X; *juEx8096* | This work | CZ28067 | ***Figure 3*** |
| Genetic reagent (*C. elegans*) | *acr-2(n2420)* X; *juEx8087* | This work | CZ28057 | ***Figure 3*** |
| Genetic reagent (*C. elegans*) | *acr-2(n2420)* X; *juEx8088* | This work | CZ28058 | ***Figure 3*** |
| Genetic reagent (*C. elegans*) | *acr-2(n2420)* X; *juEx8089* | This work | CZ28064 | ***Figure 3*** |
| Genetic reagent (*C. elegans*) | *acr-2(n2420)* X; *juEx8090* | This work | CZ28065 | ***Figure 3*** |
| Genetic reagent (*C. elegans*) | *unc-119(tm4063)* III; *wgIs433* | ***Sarov et al., 2006*** | OP433 | |
| Genetic reagent (*C. elegans*) | *eif-3.G(ju807)* II; *unc-119(tm4063)*III; *wgIs433* | This work | CZ28145 | ***Figure 6C*** |

*Continued on next page*

*Continued*

| Reagent type (species) or resource | Designation | Source or reference | Identifiers | Additional information |
|---|---|---|---|---|
| Genetic reagent (*C. elegans*) | *acr-2(n2420)* X; *unc-119(tm4063)* III; *wgIs433* | This work | CZ27913 | *Figure 6C* |
| Genetic reagent (*C. elegans*) | *eif-3.G(ju807)* II; *unc-119(tm4063)* III; *acr-2(n2420)* X; *wgIs433* | This work | CZ27914 | *Figure 6C* |
| Genetic reagent (*C. elegans*) | *sqIs17* | *Dittman and Kaplan, 2006* | MAH240 | |
| Genetic reagent (*C. elegans*) | *eif-3.G(ju807)* II; *sqIs17* | This work | CZ28334 | *Figure 6—figure supplement 1* |
| Genetic reagent (*C. elegans*) | *acr-2(n2420)* X; *sqIs17* | This work | CZ28212 | *Figure 6—figure supplement 1* |
| Genetic reagent (*C. elegans*) | *eif-3.G(ju807)* II; *acr-2(n2420)* X; *sqIs17* | This work | CZ28218 | *Figure 6—figure supplement 1* |
| Genetic reagent (*C. elegans*) | *unc-13(s69)* I; *wgIs433* | This work | CZ28491 | *Figure 6C* |
| Genetic reagent (*C. elegans*) | *unc-13(s69)* I; *acr-2(n2420)* X; *wgIs433* | This work | CZ28492 | *Figure 6C* |
| Genetic reagent (*C. elegans*) | *unc-13(s69)* I; *eif-3.G(ju807)*; *acr-2(n2420)* X; *wgIs433* | This work | CZ28493 | *Figure 6C* |
| Genetic reagent (*C. elegans*) | *juSi260 ncs-2(tm1943)* I | *Zhou et al., 2017* | CZ22459 | |
| Genetic reagent (*C. elegans*) | *juSi260 ncs-2(tm1943)* I; *eif-3.G(ju807)* II | This work | CZ23225 | *Figure 7B* |
| Genetic reagent (*C. elegans*) | *juSi260 ncs-2(tm1943)* I; *acr-2(n2420)* X | This work | CZ22345 | *Figure 7B* |
| Genetic reagent (*C. elegans*) | *juSi260 ncs-2(tm1943)* I; *eif-3.G(ju807)* II; *acr-2(n2420)* X | This work | CZ28110 | *Figure 7B* |
| Genetic reagent (*C. elegans*) | *juSi391 ncs-2(tm1943)* I | This work | CZ28213 | *Figure 7C* |
| Genetic reagent (*C. elegans*) | *juSi391 ncs-2(tm1943)* I; *eif-3.G(ju807)* II | This work | CZ28340 | *Figure 7C* |
| Genetic reagent (*C. elegans*) | *juSi391 ncs-2(tm1943)* I; *acr-2(n2420)* X | This work | CZ28252 | *Figure 7C* |
| Genetic reagent (*C. elegans*) | *juSi391 ncs-2(tm1943)* I; *eif-3.G(ju807)* II; *acr-2(n2420)* X | This work | CZ28253 | *Figure 7C* |
| Genetic reagent (*C. elegans*) | *juSi392 ncs-2(tm1943)* I | This work | CZ28277 | *Figure 7—figure supplement 1B* |
| Genetic reagent (*C. elegans*) | *juSi392 ncs-2(tm1943)* I; *eif-3.G(ju807)* II | This work | CZ28312 | *Figure 7—figure supplement 1B* |
| Genetic reagent (*C. elegans*) | *juSi392 ncs-2(tm1943)* I; *acr-2(n2420)* X | This work | CZ28291 | *Figure 7—figure supplement 1B* |
| Genetic reagent (*C. elegans*) | *juSi392 ncs-2(tm1943)* I; *eif-3.G(ju807)* II; *acr-2(n2420)* X | This work | CZ28292 | *Figure 7—figure supplement 1B* |
| Genetic reagent (*C. elegans*) | *juSi393 ncs-2(tm1943)* I | This work | CZ28278 | *Figure 7—figure supplement 1C* |
| Genetic reagent (*C. elegans*) | *juSi393 ncs-2(tm1943)* I; *eif-3.G(ju807)* II | This work | CZ28311 | *Figure 7—figure supplement 1C* |
| Genetic reagent (*C. elegans*) | *juSi393 ncs-2(tm1943)* I; *acr-2(n2420)* X | This work | CZ28293 | *Figure 7—figure supplement 1C* |
| Genetic reagent (*C. elegans*) | *juSi393 ncs-2(tm1943)* I; *eif-3.G(ju807)* II; *acr-2(n2420)* X | This work | CZ28294 | *Figure 7—figure supplement 1C* |
| Genetic reagent (*C. elegans*) | *juEx2045* | — | CZ9635 | |

*Continued on next page*

*Continued*

| Reagent type (species) or resource | Designation | Source or reference | Identifiers | Additional information |
|---|---|---|---|---|
| Genetic reagent (*C. elegans*) | *hlh-30(tm1978) IV* | CGC | CZ23321 | |
| Genetic reagent (*C. elegans*) | *hlh-30(tm1978) IV; acr-2(n2420) X* | This work | CZ28174 | Related to *Figure 6C* |
| Genetic reagent (*C. elegans*) | *eif-3.G(ju807) II; hlh-30(tm1978) IV; acr-2(n2420) X* | This work | CZ28175 | Related to *Figure 6C* |
| Genetic reagent (*C. elegans*) | *eif-3.G(ju1327) II /mnC1; juSi363 IV; acr-2(n2420) X* | This work | CZ26759 | Related to *Figure 4—figure supplement 1A* |
| Genetic reagent (*C. elegans*) | *eif-3.G(ju1327) II / mnC1 II; juSi366 IV; acr-2(n2420) X* | This work | CZ26760 | Related to *Figure 4—figure supplement 1A* |
| Genetic reagent (*C. elegans*) | *juSi364 IV; acr-2(n2420) X* | This work | CZ26494 | *Figure 4—figure supplement 1A* |
| Genetic reagent (*C. elegans*) | *eif-3.G(ju807)II juSi364 IV; acr-2(n2420) X* | This work | CZ26243 | Related to *Figure 4—figure supplement 1A* |
| Genetic reagent (*C. elegans*) | *juSi365 IV* | This work | CZ26588 | Related to *Figure 4—figure supplement 1A* |
| Genetic reagent (*C. elegans*) | *eif-3.G(ju807) II; juSi365 IV; acr-2(n2420) X* | This work | CZ26565 | *Figure 4—figure supplement 1A* |
| Genetic reagent (*C. elegans*) | *juSi365 IV; acr-2(n2420) X* | This work | CZ26566 | Related to *Figure 4—figure supplement 1A* |
| Genetic reagent (*C. elegans*) | *juSi368 IV* | This work | CZ26656 | Related to *Figure 4—figure supplement 1A* |
| Genetic reagent (*C. elegans*) | *juSi368 IV; acr-2(n2420) X* | This work | CZ26623 | *Figure 4—figure supplement 1A* |
| Genetic reagent (*C. elegans*) | *eif-3.G(ju807) II; juSi368 IV; acr-2(n2420) X* | This work | CZ26480 | Related to *Figure 4—figure supplement 1A* |
| Genetic reagent (*C. elegans*) | *wgIs506* | *Sarov et al., 2006* | OP506 | *Supplementary file 1* |
| Genetic reagent (*C. elegans*) | *acr-2(n2420) X; wgIs506* | This work | CZ27926 | *Supplementary file 1* |
| Genetic reagent (*C. elegans*) | *eif-3.G(ju807) II; acr-2(n2420) X; wgIs506* | This work | CZ27927 | *Supplementary file 1* |
| Genetic reagent (*C. elegans*) | *dhc-1::GFP(it45) I* | *Lapierre et al., 2013* | OD2955 | *Supplementary file 1* |
| Genetic reagent (*C. elegans*) | *dhc-1::GFP(it45) I; acr-2(n2420) X* | This work | CZ27858 | *Supplementary file 1* |
| Genetic reagent (*C. elegans*) | *dhc-1::GFP(it45) I; eif-3.G(ju807) II; acr-2(n2420) X* | This work | CZ27859 | *Supplementary file 1* |
| Genetic reagent (*C. elegans*) | *wgIs432* | *Sarov et al., 2006* | OP432 | *Supplementary file 1* |
| Genetic reagent (*C. elegans*) | *acr-2(n2420) X; wgIs432* | This work | CZ27915 | *Supplementary file 1* |
| Genetic reagent (*C. elegans*) | *eif-3.G(ju807) II; acr-2(n2420) X; wgIs432* | This work | CZ28021 | *Supplementary file 1* |
| Genetic reagent (*C. elegans*) | *wgIs638* | *Sarov et al., 2006* | OP638 | *Supplementary file 1* |
| Genetic reagent (*C. elegans*) | *unc-119(tm4063) III; acr-2(n2420) X; wgIs638* | This work | CZ28108 | *Supplementary file 1* |
| Genetic reagent (*C. elegans*) | *eif-3.G(ju807) II; acr-2(n2420) X; wgIs638* | This work | CZ27916 | *Supplementary file 1* |

*Continued on next page*

*Continued*

| Reagent type (species) or resource | Designation | Source or reference | Identifiers | Additional information |
|---|---|---|---|---|
| Genetic reagent (*C. elegans*) | *let-607(tm1423)* I; *unc-119(ed3)* III; *vrIs121* | *Sarov et al., 2006* | YL651 | *Supplementary file 1* |
| Genetic reagent (*C. elegans*) | *eif-3.G(ju807)* II; *let-607(tm1423)* I; *unc-119(ed3)* III; *vrIs121* | This work | CZ28143 | *Supplementary file 1* |
| Genetic reagent (*C. elegans*) | *acr-2(n2420)* X; *let-607(tm1423)* I; *unc-119(ed3)* III; *vrIs121* | This work | CZ28119 | *Supplementary file 1* |
| Genetic reagent (*C. elegans*) | *eif-3.G(ju807)* II; *acr-2(n2420)* X; *let-607(tm1423)* I; *unc-119(ed3)* III; *vrIs121* | This work | CZ28111 | *Supplementary file 1* |
| Genetic reagent (*C. elegans*) | *juIs172* | CGC | EE86 | |
| Genetic reagent (*C. elegans*) | *egl-30(md186)* I; *dpy-20(e1282ts)* IV; *syIs105* | CGC | PS4263 | |
| Genetic reagent (*C. elegans*) | *juEx7964* | *McCulloch et al., 2020* | CZ27420 | |
| Genetic reagent (*C. elegans*) | *acr-2(n2420)* X; *juEx7964* | *McCulloch et al., 2020* | CZ27217 | |
| Genetic reagent (*C. elegans*) | *eif-3.G(C130Y)* II; *acr-2(n2420)* X; *juEx7964* | This work | CZ28109 | *Supplementary file 1* |
| Recombinant DNA reagent (plasmid) | pCZGY2729 | *Andrusiak et al., 2019* | RRID:Addgene_135096 | Site-specific insertion using CRISPR/Cas9 editing of *C. elegans* ChrIV |
| Recombinant DNA reagent (plasmid) | pCZGY2750 | *Andrusiak et al., 2019* | RRID:Addgene_135094 | Expresses Cas9 and sgRNA for editing of *C. elegans* ChrIV |
| Recombinant DNA reagent (plasmid) | pCZGY2727 | This work | | Site-specific insertion using CRISPR/Cas9 editing of *C. elegans* ChrI |
| Recombinant DNA reagent (plasmid) | pCZGY2748 | This work | | Expresses Cas9 and sgRNA for editing of *C. elegans* ChrI |

### *C. elegans* genetics

All *C. elegans* strains were maintained at 20°C on nematode growth media (NGM) plates seeded with OP50 bacteria (*Brenner, 1974*). Compound mutants were generated using standard *C. elegans* genetic procedures and strain genotypes are listed in key resource table and *Supplementary file 1*. Primers for genotyping are in *Supplementary file 2*.

### Identification of *eif-3.G(ju807)*

We employed a custom workflow on the GALAXY platform to identify SNPs unique to strains containing suppressor mutations of *acr-2*(*gf*), compared to the N2 reference strain (*McCulloch et al., 2017*). Following SNP mapping using genetic recombinants, we located *ju807* to *eif-3.G* on chromosome II. We then performed transgenic expression experiments and found that both over-expression and single-copy expression of *eif-3.G*(+) in *ju807*; *acr-2*(*gf*) animals restored convulsions.

### Quantification of convulsion behavior

Convulsions were defined as contractions that briefly shorten animal body length, as previously reported (*Jospin et al., 2009*; *Video 2*). L4 larvae were cultured overnight on fresh NGM plates seeded with OP50 bacteria at 20°C. The following day, each young adult was moved to a fresh seeded plate, and after climatized for 90 s, convulsions were counted over a subsequent 90 s. The average convulsion frequency represented data over 60 s. All statistical tests were performed using GraphPad Prism6 software and p-values <0.05 were considered significant.

## CRISPR-mediated genome editing

We used a previously described method (*Dickinson et al., 2013*) with minor modifications to generate *eif-3.G(ju1327)* deletion allele. Briefly, we designed sgRNA target sequence CAATTCACAA-GAAATCGCGC, and cloned it into a Cas9-sgRNA expression construct pSK136 (derived from pDD162, with site-directed mutagenesis). A DNA mixture containing 50 ng/µl pSK136, 1 ng/µl *Pmyo-2::mCherry* (pCFJ90), and 50 ng/µl 100 bp ladder (Invitrogen, Carlsbad, CA) was microinjected into N2 adults. We screened F2 progenies from F1 animals carrying the co-injection marker for deletions in *eif-3.G* and identified a 19 bp deletion, designated *ju1327*. Heterozygous *ju1327* was twice outcrossed to N2 and then crossed to the *mnC1* balancer for stable strain maintenance (CZ22974).

The *eif-3.G(ju1840)* allele, which causes a C127Y mutation in EIF-3.G, was generated using a co-CRISPR genome editing method with *unc-58(gf)* as a selection marker (*Paix et al., 2017*). We microinjected a Cas9 complex containing the sgRNA sequence GGTCGTTTCCTTTGCAATGA, a DNA repair template incorporating TAT (encoding Y127) in place of TGC (C127), and a previously described sgRNA and repair template for *unc-58(gf)* into N2 adult hermaphrodites. We genotyped for *eif-3.G(ju1840)* among heterozygous *unc-58(gf)* F1 progeny and subsequently identified F2 animals homozygous for *eif-3.G(ju1840)* and *unc-58(+)*.

## Molecular biology and transgenesis

All transgene constructs were cloned using the Gateway cloning system (Invitrogen, Carlsbad, CA) or Gibson Assembly (NEB, Ipswich, MA), unless otherwise noted. Primers used in their construction are detailed in *Supplementary file 3*. For single-copy insertion transgenes, we used a previously described CRISPR/Cas9 method to integrate a single genomic copy on chromosome IV (*Andrusiak et al., 2019*). For extrachromosomal transgenes, we microinjected a DNA mixture containing 2 ng/µl transgene plasmid, 2.5 ng/µl pCFJ90(*Pmyo-2*::mCherry), and 50 ng/µl 100 bp ladder (Invitrogen, Carlsbad, CA) into young adults, following standard procedure (*Mello et al., 1991*).

To generate the *eif-3.G(+)* or *eif-3.G(C130Y)* genomic constructs (pCZGY3006 or pCZGY3007), we amplified a 2223 bp region from genomic DNA of N2 or CZ21759 *eif-3.G(C130Y); acr-2(gf)*, respectively, which includes 1714 bp upstream of the start codon of isoform A (F22B5.2a.1) and 331 bp downstream of the stop codon, and cloned the amplicon into the PCR8 vector (Invitrogen, CA).

To generate all *eif-3.G* cDNA expression clones, we made mixed-stage cDNA libraries with poly-dT primer for N2 or CZ21759 using Superscript III (ThermoFisher Scientific, San Diego, CA). We then amplified and *eif-3.G* cDNA using primers for the SL1 trans-splice leader (YJ74) and *eif-3.G* isoform A 3′UTR (YJ11560) and Phusion polymerase (Thermo Fisher Scientific, San Diego, CA). The cDNA clones in PCR8 vector were then used to generate tissue-specific expression constructs using Gateway cloning destination vectors (pCZGY1091 for *Punc-17β*, pCZGY925 for *Pmyo-3*, pCZGY66 for *Prgef-1*, and pCZGY80 for *Punc-25*).

We used PCR site-directed mutagenesis, in which the nucleotide changes are introduced by the primers to generate the *Prgef-1::eif-3.G(ΔRRM)* and *Prgef-1::eif-3.G(C130Y ΔRRM)* constructs (pCZGY3026 and pCZGY3027, respectively) with primers YJ11561 and YJ11562 on the templates pCZGY2715 and pCZGY2716, respectively. The *Prgef-1::eif-3.G(RFF/AAA)* construct (pCZGY3512) was generated by two rounds site directed PCR mutagenesis on pCZGY3010, first using primers YJ12463 and YJ12464, then primers YJ12465 and YJ12466. To generate *Pref-1::eif-3.I(+)* (pCZGY3508), we amplified *eif-3.I* cDNA from N2 cDNA libraries using primers YJ12453 and YJ12454, and used Gibson Assembly to clone into the pCZGY66 backbone containing *Prgef-1*. We then performed site-directed mutagenesis on pCZGY3508 using primers YJ12457 and YJ12458 to generate the *Prgef-1::eif-3.I(Q252R)* construct (pCZGY3509).

We generated the GFP::EIF-3.G clones pCZGY3018 and pCZGY3019 via Gibson assembly, using *eif-3.G(+)* or *eif-3.G(C130Y)* cDNA amplified using primers YJ12604 and YJ12605, and the GFP-coding DNA amplified using primers YJ12602 and YJ12603.

To generate *Punc-17β*::EIF-3.G::3xFLAG::SL2::GFP constructs (pCZGY3538 for WT, pCZGY3539 for C130Y, and pCZGY3540 for ΔRRM) used in seCLIP experiments, *Punc-17β* promoter was amplified from pCZGY1091 using primers YJ12164 and YJ12418, each *eif-3.G* cDNA (wild type, C130Y, or ΔRRM) was amplified with an N-terminal 3xFLAG sequence from subclones using the primers YJ12419 and YJ12420, SL2 trans-splice sequence was amplified from N2 genomic DNA using primers YJ12421 and YJ12422, and GFP was amplified from pCZGY3018 using primers YJ12423 and

YJ12424. These fragments were then Gibson Assembled into the pCZGY2729 backbone (RRID: Addgene_135096), which facilitates CRISPR/Cas9 single copy insertion on chromosome IV (*Andrusiak et al., 2019*).

All *ncs-2* transgenes were similarly cloned using primers for Gibson assembly into pCZGY2727. To generate the *Pncs-2*::5′UTR mutant::*ncs-2* cDNA construct (pCZGY3526), we amplified *Pncs-2* from N2 gDNA using primers YJ12554 and YJ12555. A fragment containing SL1 trans-spliced *eif-3.G* 5′UTR incorporated in the forward primer, *ncs-2* cDNA, GFP, and the *ncs-2* 3′UTR was amplified from CZ22459 gDNA using primers YJ12556 and YJ12557. The *Pncs-2*::GFP(+) construct (pCZGY3533) was cloned by amplifying *Pncs-2* through the first four codons of *ncs-2* CDS from N2 gDNA using primers YJ12554 and YJ12579, and GFP and the *ncs-2* 3′UTR from CZ22459 gDNA using YJ12580 and YJ12557. The *Pncs-2*::5′UTR mutant::GFP construct (pCZGY3534) was cloned by amplifying *Pncs-2* from N2 gDNA using primers YJ12554 and YJ12555, and the *eif-3.G* 5′UTR, the first four codons of the *ncs-2* CDS, GFP, and the *ncs-2* 3′UTR from CZ22459 gDNA using primers YJ12581 and YJ12557.

## Fluorescence microscopy and GFP intensity quantification

L4 or young adult animals were immobilized in 1 mM levamisole in M9 and mounted on microscope slides with 2% agar. All images were collected on a Zeiss LSM800 confocal microscope, unless specified, with identical image acquisition settings: 1.25 μm pixel size with 0.76 μs pixel time, 50 μm pinhole, with genotype-blinding to observer when possible. The positions of VA10, VB11, and DB7 cholinergic motor neurons were identified using *juEx2045*(*Pacr-2-mCherry*), based on their stereotypical patterning in the posterior ventral nerve cord. These neurons were chosen for quantification because they were consistently visible in single focal plane images. All quantification of GFP intensity in these neurons was performed using the Integrated Density function in ImageJ (*Schindelin et al., 2012*). We acquired the mean integrated density from the VA10, VB11, and DB7 cell bodies, subtracted background intensity from an equivalent area, and the resulting values were then normalized to the mean area of the cell bodies of the same animal. We similarly quantified fluorescence intensities in the ventral nerve cord of animals expressing GFP-tagged full-length *ncs-2* cDNA, except integrated densities were obtained from one ROI per image (red boxes in *Figure 7B and C*). All data was normalized to the mean fluorescence intensity of the transgene in the wildtype background. All statistical analysis was performed with GraphPad Prism6 software.

Axon commissures, observed as fluorescent structures extending from the ventrally located neuron cell body to the dorsal body wall, shown in *Figure 1—figure supplement 2A* were visualized with *juIs14*[*Pacr-2*::GFP] and manually quantified. Imaging shown in *Figure 1—figure supplement 2B* was performed using a Zeiss Axioplan two microscope installed with Chroma HQ filters and a 63x objective lens. Synaptic puncta labeled by *nuIs94*[SNB-1::GFP], were manually quantified in the region anterior to the ventral nerve chord between VD6 and VD7.

## Polysome profiling

We prepared *C. elegans* lysates and sucrose gradients using the protocol described in *Ding and Grosshans, 2009*. To synchronize animals, gravitated adults were treated with 20% Alkaline Hypochlorite Solution and embryos were plated on four 30 cm NGM plates seeded with OP50, and grown to the L4 stage at 20°C. Approximately 200 μl packed L4 *C. elegans* were harvested by centrifugation in M9 media at 1500 RPM, washed three times in ice-cold M9 media supplemented with 1 mM cycloheximide, then once more in lysis buffer base solution (140 mM KCl, 20 mM Tris-HCl (pH 8.5), 1.5 mM MgCl$_2$, 0.5% NP-40, 1 mM DTT, 1 mM cycloheximide) followed by snap freezing in liquid nitrogen. The frozen pellets were resuspended in 450 μl lysis buffer (140 mM KCl, 20 mM Tris-HCl (pH 8.5), 1.5 mM MgCl2, 0.5% NP-40, 2% PTE, 1% sodium deoxycholate, 1 mM DTT, 1 mM cycloheximide, 0.4 units/μl RNAsin) and crushed to a fine powder with a mortar and pestle pre-cooled with liquid nitrogen. Protein lysate concentrations were then determined using a Bradford assay (Bio-Rad, Hercules, CA). Fifteen to 60% sucrose gradients were prepared in 89 mm polypropylene centrifuge tubes (Beckman Coulter) using standard settings on a Foxy Jr. density gradient fractionation system (Teledyne ISCO, Lincoln, NE) and lysate volumes corresponding to equal protein amounts between samples were loaded on top of the gradients. Loaded gradients were then spun in an Optima L-80 ultracentrifuge (Beckman Coulter) at 36,000 rpm at 4°C for 3 hr. Fractions were

then collected and RNA absorbance was continuously acquired using a UA-6 detector (Teledyne ISCO, Lincoln, NE) with a 70% sucrose chase solution. We calculated the area under the curve (AUC) for monosome (80S) and polysome absorbance traces using the Simpson's rule method in SciPy (*Virtanen et al., 2020*) and used the AUC values to calculate the polysome to monosome ratios.

### Western blot analysis

A total of 500 µl of mixed staged worms were resuspended in lysis buffer (140 mM KCl, 20 mM Tris-HCl (pH 8.5), 1.5 mM MgCl2, 0.5% NP-40, 1% sodium deoxycholate, 1 mM DTT) supplemented with protease inhibitors (Complete Ultra Tablets, Roche), frozen in liquid nitrogen, and crushed to a fine powder. The lysates were clarified by centrifugation at max speed in a tabletop centrifuge and protein levels were quantified using a Bradford assay (Bio-Rad, Hercules, CA). The resulting protein lysates were then boiled in Laemmli buffer with 10% 2-mercaptoethanol, run on SDS-PAGE gels, and transferred to PVDF blots, which were probed with anti-FLAG (F7425, RRID:AB_439687) or anti-Actin (clone C4, RRID:AB_2335304) antibodies.

### seCLIP library preparation and sequencing

We performed single-end enhanced CrossLink and ImmunoPrecipitation (seCLIP) experiments according to the published protocol in *Van Nostrand et al., 2017*, with the following adjustments to ensure efficient immunoprecipitation yield from *C. elegans* lysates. Mixed stage animals were grown on ~12 NGM plates (30 cm) and washed twice with M9, spinning at 1500 rpm between washes. Animals were then resuspended in 5 ml M9 media and rocked on a rotator for 10 min to remove gut bacteria, followed by one more wash with M9 at 1500 rpm. The animals were spread on one NGM plate (30 cm) and then UV-crosslinked with a Spectrolinker XL-1000 (Spectronics, New Cassel, NY) using energy setting 3 kJ/m$^2$ according to *Broughton and Pasquinelli, 2013*. Afterwards, animals were resuspended in 4 ml lysis buffer [150 mM NaCl, 1 M HEPES, 100 mM DTT, 6.25 µl RNAsin (Promega) per 10 ml, 10% glycerol, 10% Triton X-100, one protease inhibitor tablet per 10 ml] and split into two tubes for each replicate. The resuspension was disrupted on an XL-2000 Sonicator (QSonica, Newtown, CT) with seven pulses (powersetting = 11, 10 s each, 50 s on ice in between) and immediately spun at 4750 RPM for 5 min at 4°C. All subsequent steps, beginning with RNAse A treatment of the supernatant, was performed according to the seCLIP protocol (*Van Nostrand et al., 2017*), except that high-salt and low-salt wash buffers were replaced with a single buffer (2M NaCl, 1M HEPES, 30% glycerol, 1% Triton X-100, one protease inhibitor tablet per 10 mL) optimized for anti-FLAG RNA IP from *C. elegans* lysates (*Blazie et al., 2015*). Immunoprecipitation was performed with anti-FLAG beads (Sigma, RRID:AB_2637089). cDNA libraries were prepared from both the immunoprecipitated mRNA (CLIP) as well as the sample before immunoprecipitation (INPUT), such that crosslink sites can be defined by read enrichment in the CLIP sample over input as described (*Van Nostrand et al., 2017*). seCLIP libraries were validated using the D1000 high sensitivity screen tape system (Agilent, La Jolla, CA) and quantified using a Qubit instrument (Thermo Fisher, San Diego, CA) before pooling and sequencing on HiSeq4000 (Illumina, San Diego, CA) at the IGM Genomics Center, University of California San Diego.

### seCLIP sequence mapping

After demultiplexing barcoded reads, we used the CLIPPER software pipeline (*Lovci et al., 2013*) to trim barcodes, remove PCR duplicate reads, filter reads mapping to repetitive elements, and map the remaining reads to the *C. elegans* reference genome (ce10). The total number of uniquely mapped reads obtained after filtering is in *Supplementary file 4*. A large proportion of reads obtained from the ΔRRM and IgG samples mapped to repetitive elements and were discarded, explaining the smaller number of uniquely mapped reads in these samples. In seCLIP, RNA-binding sites are defined as read clusters enriched in the crosslink immunoprecipitated sample (CLIP) over the input control (INPUT) (*Van Nostrand et al., 2017*), which are comprehensively identified across each dataset using CLIPPER. Read clusters were reproducibly identified from independent biological replicates of seCLIP, except in the ΔRRM control reflecting background, supporting the specificity of our data (*Figure 4—figure supplement 1*).

## EIF-3.G footprint identification from mapped seCLIP reads

We defined EIF-3.G footprints as seCLIP read clusters appearing in both replicates with 20 reads and 1.5 fold-change enrichment over the INPUT control in at least one replicate. Footprints matching these criteria in the IgG (no transgene) and the EIF-3.G(ΔRRM) control samples were considered background and subtracted from the EIF-3.G(WT) and EIF-3.G(C130Y) datasets (*Supplementary files 5* and *6*). We annotated footprints to their gene features (eg. 5′UTR, CDS) using a script (*Yee, 2021*; https://github.com/byee4/annotator) that overlaps read clusters with the *C. elegans* genome annotation WS235. We grouped all clusters annotated in the CDS and 5′UTR into one category (5′UTR proximal), since clusters mapping in CDS were almost always located within 200nts of a 5′UTR (*Figure 4D*).

## Gene ontology and KEGG pathway analysis

GO analysis was performed using 225 EIF-3.G target genes as input to the biological process annotation set within the Gene Ontology Resource tool (*Ashburner et al., 2000*). A total 211 gene names were recognized by the database and GO term enrichment was defined using a threshold of $p < 0.05$. Pathway analysis of EIF-3.G target genes was performed using the Kyoto Encyclopedia of Genes and Genomes (KEGG) annotation tool within the DAVID bioinformatics resource (*Jiao et al., 2012*) using default settings.

## Analysis of activity-dependent expression changes among EIF-3.G target mRNAs in cholinergic neurons

We studied activity-dependent transcript expression changes among the EIF-3.G target genes (n = 225) by re-analyzing the cholinergic neuron-specific transcriptomes reported in *McCulloch et al., 2020* using the Galaxy platform (*Afgan et al., 2018*). We downloaded raw FASTA reads from transcriptome sequencing of wild type and *acr-2*(*gf*) animals (n = two replicates each; accession #'s SRR10320705, SRR10320706, SRR10320707, SRR10320707) and mapped them to the *C. elegans* reference genome (ce10) using BWA (*Li and Durbin, 2009*). Differential expression among the EIF-3.G target genes was quantified using Feature Counts (*Liao et al., 2014*) and DeSeq2 (*Love et al., 2014*).

## in silico analysis of 5′UTR sequence features, secondary structure, and conservation

We downloaded all *C. elegans* transcript 5′UTRs (WS271) from Parasite Biomart (*Howe et al., 2016*), and calculated 5′UTR lengths as the sequence between the 5′ distal end and the start codon of each transcript. To have meaningful length calculation, we only considered 5′UTRs annotated with at least 10nt and restricted our analysis to the longest 5′UTR isoform for each gene to avoid considering multiple transcripts of the same gene. By these criteria we identified 5′UTRs for 10,962 WS271 protein coding transcripts and 179 transcripts with EIF-3.G footprints. We used the same criteria to determine features of the *acr-2*(*gf*) cholinergic transcriptome 5′UTRs (*McCulloch et al., 2020*) for the analysis shown in *Figure 4E–F*.

For the analysis shown in *Figure 4—figure supplement 2D–E*, the genomic coordinates of human gene 5′UTRs were downloaded from Ensembl and used to obtain 5′UTR sequences from the human genome reference sequence (hg38). eIF3g footprints from HEK293 cells were previously described (*Lee et al., 2015*). We defined our analysis of 5′UTR sequences using the same criteria described for *C. elegans* and the data show the comparison between 5′UTRs of 255 genes with human eIF3g footprints and 19,914 total genes in the human genome annotation (hg38).

To calculate GC-enrichment, we used BEDTools (*Quinlan and Hall, 2010*) to generate a FASTA of 5′UTR sequences from their genomic coordinates and used Biopython (*Cock et al., 2009*; https://github.com/biopython/biopython) to calculate the total %GC in their sequences (*Figure 4F*) as well as %GC within 10nt bins incremented from the start codon ATG for the analysis shown in *Figure 4—figure supplement 2C*.

To predict secondary structures of the 5′ ends of *hlh-30d*, *ncs-2*, and *eif-3.G* mRNAs, we used the RNAfold Web Server (*Gruber et al., 2008*) with default settings. To better understand the contribution of gene-specific 5′UTR sequences, we excluded the SL1 sequences of *ncs-2* and *eif-3.G* from folding predictions. The free energies (ΔG) for each sequence reported in our results were derived

from the reported thermodynamic ensemble. Data showing conservation of *eif-3.G* and *ncs-2* 5′UTR sequences compared with 135 nematode species (phyloP135way scores) was obtained from the UCSC Genome Browser with the genomic position along the sequence of each 5′UTR as input.

## Acknowledgements

We thank Brian Yee, Eric Van Nostrand, Gabriel Pratt, and Gene Yeo for advice on adapting the seCLIP protocol to *C. elegans* and technical support with seCLIP data analysis; Timothy Shaw and Jens Lykke-Andersen for sharing equipment and invaluable assistance with polysome profile analysis; Yan Zhao, Ann Zhou and Ippei Ozaki for assistance in constructing expression clones and strains; Yunbo Li for advice on protein analysis; Malena Hansen, Josh Kaplan, and Keming Zhou for transgenes, Kenneth Miller for P*unc-17β* plasmid. Some strains used in this study were provided by the Caenorhabditis Genetics Center, which is funded by NIH Office of Research Infrastructure Programs (P40 OD010440). We acknowledge WormBase and UCSC Genome databases for genomic information resource. We thank Andrew D Chisholm and members of our laboratory for critical reading of the manuscript. S B was a trainee on the UCSD T32 training grant (NS007220). This work is funded by NIH grant (NS R37 035546 to Y J).

## Additional information

### Funding

| Funder | Grant reference number | Author |
|---|---|---|
| National Institutes of Health | NS R37 035546 | Yishi Jin |
| University of California, San Diego | NS007220 | Stephen M Blazie |

The funders had no role in study design, data collection and interpretation, or the decision to submit the work for publication.

### Author contributions

Stephen M Blazie, Conceptualization, Data curation, Formal analysis, Investigation, Methodology, Writing - original draft, Writing - review and editing; Seika Takayanagi-Kiya, Conceptualization, Data curation, Formal analysis, Investigation, Methodology, Writing - review and editing; Katherine A McCulloch, Data curation, Writing - review and editing; Yishi Jin, Conceptualization, Resources, Supervision, Funding acquisition, Project administration, Writing - review and editing

### Author ORCIDs

Stephen M Blazie ORCID https://orcid.org/0000-0001-6701-6275
Yishi Jin ORCID https://orcid.org/0000-0002-9371-9860

### Decision letter and Author response

Decision letter https://doi.org/10.7554/eLife.68336.sa1
Author response https://doi.org/10.7554/eLife.68336.sa2

## Additional files

### Supplementary files

- Supplementary file 1. Strains used in this study.

- Supplementary file 2. Genotyping primers used in this study.

- Supplementary file 3. Constructs and related primers used in this study.

- Supplementary file 4. Number of mapped reads in seCLIP replicate datasets obtained after sequencing and CLIPPER filtering.

• Supplementary file 5. Number of read clusters detected in each dataset after subtraction of IgG control background.

• Supplementary file 6. Number of EIF-3.G footprints detected in each dataset after subtraction of background from both IgG and ΔRRM controls.

• Transparent reporting form

## Data availability

Raw and processed seCLIP datasets from this study have been uploaded to the Gene Expression Omnibus (GEO) under accession number GSE152704.

The following dataset was generated:

| Author(s) | Year | Dataset title | Dataset URL | Database and Identifier |
|---|---|---|---|---|
| Blazie SM, Takayanagi-Kiya S, McCulloch KA, Jin Y | 2021 | seCLIP of *C. elegans* EIF-3.G in the cholinergic motor neurons | https://www.ncbi.nlm. nih.gov/geo/query/acc. cgi?acc=GSE152704 | NCBI Gene Expression Omnibus, GSE152704 |

The following previously published dataset was used:

| Author(s) | Year | Dataset title | Dataset URL | Database and Identifier |
|---|---|---|---|---|
| McCulloch KA, Zhou K, Jin Y | 2019 | Neuronal transcriptome analyses reveal novel neuropeptide modulators of excitation and inhibition imbalance in *C. elegans* | https://www.ncbi.nlm. nih.gov/geo/query/acc. cgi?acc=GSE139212 | NCBI Gene Expression Omnibus, GSE139212 |

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
