## [Decision Letter]

**Acceptance summary:**

This study takes advantage of unbiased genetics to discover functions of EIF-3.G in neuronal protein synthesis. This is especially interesting and timely given the connection of the eIF3 complex with various diseases of the nervous system and a previous lack of detailed understanding of how EIF-3.G specifically plays a role in translation control.

**Decision letter after peer review:**

Thank you for submitting your article "Eukaryotic initiation factor EIF-3.G augments mRNA translation efficiency to regulate neuronal activity" for consideration by *eLife*. Your article has been reviewed by 3 peer reviewers, and the evaluation has been overseen by a Reviewing Editor and Piali Sengupta as the Senior Editor. The reviewers have opted to remain anonymous.

Essential revisions:

1) The ∆RRM construct, which is a deletion, may have secondary effects on folding/stability of the encoded protein. Given that this EIF-3.G variant is used as an important control for seCLIP studies, the reviewers recommend that the authors confirm the RRM deletion mutant protein is at least expressed at similar levels as the EIF-3.G wild type and C130Y variants.

2) The reviewers encourage the authors to provide data ensuring the phenotype from the forward screen is due to the eif-3.g mutation either by i) using CRISPR to engineer a revertant of the C130Y eif-3.g mutation and asking whether this abolishes the phenotype, or alternately, ii) using the GFP tagged single copy insertion C130Y expressing strain in combination with any CRISPR mutant that disrupts the native eif-3.g locus.

3) The reviewers felt the CLIP data were an important part of the story and debated whether or not additional validations of these findings were needed. In the end the reviewers request only that the authors state whether additional GFP reporters in addition to hlh-30 and ncs-2 were tested to give the readers a sense of the potential false positive rate.

4) All of the reviewers thought that the following experimental suggestion would substantially strengthen the paper if the strains were easily available for testing. However if this study would substantially delay the publication of this work, it was not deemed essential. "It is interesting that eif-3.g(C130Y) decreases translation of hlh-30 and ncs-2 only in acr-2(gf) mutants. Is this a direct consequence of elevated neural activity in Ach MNs? For example, does silencing of these neurons attenuate the ability of eif-3.g(C130Y) to decrease ncs-2 GFP reporter translation?"

5) The reviewers had many questions about the ife-1 section of the study and felt that it was underdeveloped with respect to the rest of the paper. They suggest the authors consider removing this work from the current study and develop it further for a future publication.

6) All of the reviewers comments have been included below to provide feedback to the authors. However, only the 5 points above were determined by the consensus of the reviewers to be the major points that required addressing in the revision.

*Reviewer #1:*

In general, I found this study to be both interesting and timely given the connection of the eIF3 complex with various diseases of the nervous system and a lack of detailed understanding of how EIF-3.G specifically plays a role in translation control. Moreover, the experiments are generally sound, rigorous, and conclusions made from the interpretation of the data are reasonable and not over-stated.

Strengths:

– The genetic screen identifying the mutation in the Zinc Finger domain provided the basis for a new line of inquiry into the role of this domain in translational control.

– The first use of the single-end enhanced CLIP protocol in *C. elegans*, which will be highly useful for those studying RNA binding proteins in this model organism

– Initial insights into the features associated with EIF-3.G regulation of translation and its link to target mRNAs that themselves control neuronal activity.

Weaknesses:

– Despite initial insights into the possible mechanism of action, it is still unclear how the mutation in the Zinc Finger domain or the Zinc Finger domain itself is acting to modulate translation of specific mRNAs.

– The authors propose a model that structured 5'UTRs are likely playing a key role in the regulation of translation by EIF-3.G, but further experiments would be required to more strongly confirm this model.

Overall, I found the quality of the science and the data presented to be very high. I have only two primary suggestions for experiments that I feel would strengthen the conclusions of the current manuscript.

1) The ∆RRM construct, which is a deletion, may have secondary effects on folding/stability of the encoded protein. Given that this EIF-3.G variant is used as an important control for seCLIP studies, I think it would be useful to check that this RRM deletion mutant protein is at least expressed at similar levels as the EIF-3.G wild type and C130Y variants. Since the authors already have FLAG-tagged versions of all three proteins for their CLIP experiments, it should not be too much work to perform a Western blot to compare protein levels.

2) The connection to structured 5'UTRs is intriguing but could be further strengthened by an experiment that more explicitly tests the role of secondary structure. For example, in addition to the UTR swapping experiments performed by the authors, it would also be informative to introduce mutations at nucleotides engaged in base-pairing interactions to see if eliminating these interactions influences regulation. If effects are observed, subsequent compensatory mutations that re-introduce base-pairing should be performed to revert back to patterns observed with the native 5'UTR.

*Reviewer #2:*

In this manuscript, Blazie et al. examine the neuronal function of EIF-3.G, a RNA-binding subunit of the translation initiation complex eIF3. First, they recover a semi-dominant gain-of-function mutation in eif-3.g (a C130Y point mutation in its zinc finger domain) in a screen for suppressors of cholinergic motor neuron hyperactivity. The authors show that the eif-3.g(C130Y) phenotype can be reversed by expression of wild-type eif-3.g in Ach motor neurons and that the WT and C130Y mutant express at equal levels. The authors then show that loss of ife-1 suppresses the eif-3.g(c130y) effect and that eif-3.g's function vitally depends on its RRM domain. They then use transgenic expression of Flag-tagged-eif-3.g to profile eif-3.g binding to mRNAs in the Ach motor neurons. This reveals binding of eif-3.g to 5'UTR-proximal regions of mRNAs, particular in long, GC-rich 5'UTRs. The authors note that eif-3.g-bound mRNAs include several that show acr-2(gf)-dependent transcription, with an enrichment of genes involved in neuropeptide signaling and other categories. They then focus on two specific targets of eif-3.g. First, they show that hlh-30 translation is increased by acr-2(gf), which can be suppressed by eif-3.g(c130y). Second, they show that ncs-2 translation is unaffected by acr-2(gf), but reduced by eif-3.g(c130y) in an acr-2(gf) background.

This paper takes advantage of unbiased genetics to determine a function for eif-3.g in neuronal protein synthesis regulation. Its strengths are that it develops a new system in which eif-3.g can be studied, identifies neuronal targets of eif-3.g, and demonstrates a neuronal function for this important translational regulator. Its weaknesses are that it doesn't fully resolve how eif-3.g regulates Ach motor neuron excitability or bind together an understanding of how eif-3.g regulates its targets in a manner that depends on acr-2(gf).

In this manuscript, Blazie et al. examine the neuronal function of EIF-3.G, a RNA-binding subunit of the translation initiation complex eIF3. First, they recover a semi-dominant gain-of-function mutation in eif-3.g (a C130Y point mutation in its zinc finger domain) in a screen for suppressors of cholinergic motor neuron hyperactivity. The authors show that the eif-3.g(C130Y) phenotype can be reversed by expression of wild-type eif-3.g in Ach motor neurons and that the WT and C130Y mutant express at equal levels. The authors then show that loss of ife-1 suppresses the eif-3.g(c130y) effect and that eif-3.g's function vitally depends on its RRM domain. They then use transgenic expression of Flag-tagged-eif-3.g to profile eif-3.g binding to mRNAs in the Ach motor neurons. This reveals binding of eif-3.g to 5'UTR-proximal regions of mRNAs, particular in long, GC-rich 5'UTRs. The authors note that eif-3.g-bound mRNAs include several that show acr-2(gf)-dependent transcription, with an enrichment of genes involved in neuropeptide signaling and other categories. They then focus on two specific targets of eif-3.g. First, they show that hlh-30 translation is increased by acr-2(gf), which can be suppressed by eif-3.g(c130y). Second, they show that ncs-2 translation is unaffected by acr-2(gf), but reduced by eif-3.g(c130y) in an acr-2(gf) background.

This paper takes advantage of unbiased genetics to determine a function for eif-3.g in neuronal protein synthesis regulation. Its strengths are that it develops a new system in which eif-3.g can be studied, identifies neuronal targets of eif-3.g, and demonstrates a neuronal function for this important translational regulator. Its weaknesses are that it doesn't fully resolve how eif-3.g regulates Ach motor neuron excitability or bind together an understanding of how eif-3.g regulates its targets in a manner that depends on acr-2(gf).

1) The study provides a characterization of a new eif-3.g allele, which is interesting. However, the study does not come full circle and clarify how the eif-3.g mutation alters cholinergic neuron excitability. Nor does it show in a mechanistic sense how eif-3.g function depends on neuronal activity. It would be helpful if the authors could point to any efforts that they've made to revert the phenotype of eif-3.g via overexpression of eif-3.g(c130y)-downregulated target genes. If they were able to show such an effect, this would greatly enhance the study.

2) It is interesting that eif-3.g(C130Y) decreases translation of hlh-30 and ncs-2 only in acr-2(gf) mutants. Is this a direct consequence of elevated neural activity in Ach MNs? For example, does silencing of these neurons attenuate the ability of eif-3.g(C130Y) to decrease ncs-2 GFP reporter translation?

3) A couple of extra genetic controls would bolster the paper: (i) using CRISPR to engineer a revertant of the C130Y eif-3.g mutation and asking whether this abolishes the phenotype; and (ii) overexpressing eif-3.g(C130Y) in Ach MNs and asking whether this is sufficient to induce the phenotype.

4) The use of neuronal cell type-specific seCLIP is exciting, but only 2 replicates were conducted and there is no systematic validation of the detected binding sites. RT-qPCR should be used to validate detected binding sites on a handful of targets.

5) For Figure 4E-F and related analyses: the authors are comparing the eif-3.g targets to the full transcriptome to determine notable properties of the eif-3.g-bound gene set. I believe that for these analyses it would be more appropriate to compare the eif-3.g-bound gene set to the set of ACH MN-expressed genes (which the authors have measured in a previous study). In its current form, it is unclear whether the length and GC content of the eif-3.g-bound genes is a general property of cholinergic neuron gene expression. The same idea also applies to the GO analyses.

6) Related to the length of the eif-3.g-bound 5'UTRs, longer RNA sequences have a higher probability of having a detected binding site (i.e. the number of detected binding sites along any region of DNA/RNA should scale with length). It is not clear whether the authors correct for this in their analysis.

*Reviewer #3:*

The authors demonstrate the cholinergic motor neurons (ACh-MN)-specific function of EIF-3.G in a previously well-developed *C. elegans* model of acetylcholine receptor-2 gain-of function [acr-2(gf)], showing spontaneous seizure-like convulsions. C130Y mutation of the zinc finger domain in EIF-3.G ameliorates hyperactivity induced by acr-2(gf), while it does not affect global translation, neuronal development, and synaptic formation. EIF-3.G preferentially associates with long and GC-enriched 5'UTR genes involved in the regulation of ACh-MN synaptic activity and modulates neuronal protein synthesis in an activity-dependent manner.

The authors provide strong genetic evidence that the eif-3. g(C130Y) allele suppresses the convulsion phenotype in acr-2(gf) in a semi-dominant, gain-of-function manner in the ACh-MN neuron. The analyses using homozygous and heterozygous forms of the allele, as well as transgenic rescue experiments were expertly performed, and the data were well described. This study documents a comprehensive and detailed insight into neuronal activity-dependent mRNA translation using diverse approaches

1. The statistical representation in Figure 1E is somewhat misguided. In order to show that eif3.G increases convulsion in acr-2 (gf); eif-3.G(C130Y) in an ACh-MN-specific manner, the statistical analysis of the right 4 samples should be done with WT or (-), not with Ex[eif-3.G(+)].

2. The statement 'Both GFP::EIF-3.G(WT) and EIF-3G(C130Y) showed cytoplasmic fluorescence throughout somatic cells and across all developmental stages' is not well-presented in Figure 1—figure supplement 1B, since only the wild-type and one developmental stage (L4) was shown. It is also unclear if the bright puncta across the animal actually correspond to Peif-3.g::GFP or is a result of autofluorescence from the gut. It will be useful to include arrows and insets to highlight the pattern of interest (e.g., cytoplasmic, specific somatic tissues). To elaborate on the tissue specificity suggested in Figure 1E, whether EIF-3.G WT and C130Y are expressed in ACh-MN neuron should be discussed – This will differentiate whether the C130Y effect is cell-autonomous or non-autonomous. The authors also referred to Figure supplement 1B and Figure 2B as assessments of EIF-3.G protein stability – Expression pattern/localization would be more appropriate than protein stability here.

3. In Figure 3A, the potential involvement of ife-1 (one of the eIF4E homologs) in eif-3.g(C130Y) suppression of acr-2(gf) is interesting. Some experiments are needed to discern the direct/indirect nature of this genetic interaction.

A. Does ife-1(0) cause convulsion in eif-3.G(C130Y) alone (acr-2 wild-type)?

B. Does ife-1(0) alter eif-3.G WT and C130Y expression?

C. ife-1 was shown to be predominantly expressed in the germline where it is important for germ cell development (Henderson et al., Journal of Cell Science 2009; Amiri et al., Development 2001). Moreover, the protein is undetectable by western blot from germline-deficient glp-4(bn2) animal (Amiri et al. 2001), potentially indicating exclusive germline expression. Thus, it will be important to look at whether IFE-1 is co-expressed with EIF-3.G WT and C130Y in the soma and ACh-MN.

4. EIF-3.G preference for long 5' UTR with high GC content was nicely dissected from the seCLIP data in Figure 4. For consistency, the authors could provide information of 5'UTR length and GC content between upregulated and downregulated genes in acr-2(gf) over WT, shown in Figure 5B. hlh-30 and ncs-2 were further linked to eif-3.g-mediated suppression of acr-2(gf) by examining their expression using GFP reporters in Figures 6 and 7. Polysome profiling can be done to differentiate the effect of C130Y on the translation efficiency of these transcripts, as opposed to their transcription and protein stability. It would also be informative to show the preference of these genes for *C. elegans* eIF4E homologs, IFE-1, IFE-2, and IFE-4.

5. A heatmap in Figure 4—figure supplement 2C needs hierarchical clustering, even though genes in the map are already clustered top to bottom by increasing positional GC-density near the start codon. It may help extract additional meaningful information. The authors also need to point out the genes described in the text (pmt-2, let-607, unc-43, and gsa-1).

6. The statistical representation in the graph in Figure 6D needs to be corrected.

---

## [Author Response]

Essential revisions:1) The ∆RRM construct, which is a deletion, may have secondary effects on folding/stability of the encoded protein. Given that this EIF-3.G variant is used as an important control for seCLIP studies, the reviewers recommend that the authors confirm the RRM deletion mutant protein is at least expressed at similar levels as the EIF-3.G wild type and C130Y variants.

We appreciate the reviewers’ comment. As suggested by Reviewer #1, we carried out western blot analysis on protein extracts made from the three transgenic lines. We present this data in the revised Figure 4—figure supplement 1A, which shows that the EIF-3.G(∆RRM) truncated protein is expressed, but at reduced levels compared to EIF-3.G(WT) and EIF-3.G(C130Y). It is important to note that during seCLIP we immunoprecipitated the truncated EIF-3.G(∆RRM) protein at the same level as full-length EIF-3.G(WT) and EIF-3.G(C130Y) proteins. We obtained nearly 3 times more reads from seCLIP of 3XFLAG::EIF-3.G(∆RRM) animals (543,913 reads) than from N2 animals that lack any transgene (192,259 reads; IgG(-) negative control in Supplementary File 4).

Together, these data support that the 3XFLAG::EIF-3.G(∆RRM) transgene is expressed and serves as an effective reagent to detect RNAs specifically bound by the RRM of EIF-3.G(WT and C130Y).

We describe the results in the revised results (lines 220-226) and methods (lines 652-661).

2) The reviewers encourage the authors to provide data ensuring the phenotype from the forward screen is due to the eif-3.g mutation either by i) using CRISPR to engineer a revertant of the C130Y eif-3.g mutation and asking whether this abolishes the phenotype, or alternately, ii) using the GFP tagged single copy insertion C130Y expressing strain in combination with any CRISPR mutant that disrupts the native eif-3.g locus.

We appreciate the suggestion. While our attempt to edit C130Y back to wild type EIF-3.G was not successful, likely due to inefficiency of sgRNAs, we generated animals expressing the GFP-tagged EIF-3.G(C130Y) single copy transgene in a *eif-3.G*(*0*) background (following the suggestion ii). We found that GFP::EIF-3.G(C130Y) rescued the early larval arrest phenotype of *eif-3.G*(*0*), supporting its functionality, and additionally suppressed *acr-2*(*gf*) convulsion behavior at levels similar to endogenous *eif-3.G*(*C130Y*). The new results are shown in Figure 2A and described in the revised text (lines 158-163).

We performed an additional experiment to change the first cysteine (C127) of the zinc finger CCHC motif to tyrosine and found that *eif-3.G*(*C127Y*) behavior was identical to *eif-3.G*(*C130Y*). This result is shown in revised Figure 1 and discussed in the text (lines 107-113, 432-434, and 528-535).

Together, these results strengthen our conclusion that the suppression of *acr2(gf)* by *eif-3.g(C130Y)* isolated from the forward screen is due to the C130Y mutation in *eif-3.G*, and the integrity of the Zinc Finger domain is important for EIF-3.G function.

3) The reviewers felt the CLIP data were an important part of the story and debated whether or not additional validations of these findings were needed. In the end the reviewers request only that the authors state whether additional GFP reporters in addition to hlh-30 and ncs-2 were tested to give the readers a sense of the potential false positive rate.

We agree with the comment, and have revised our manuscript to include additional details on how our test of other GFP reporters led us to focus on *hlh-30* and *ncs-2*. We screened ten GFP reporter lines and found six lines with detectable expression in ACh-MNs. In addition to NCS-2 and HLH-30, we characterized effects of *acr-2*(*gf*) and *eif-3.G*(*C130Y*) on four other reporter lines (for genes ZIP-2, ATF-7, LET607, and FLP-12). We have now documented these observations in the revised Supplementary File 1 and in lines 325-328 of the revised text.

4) All of the reviewers thought that the following experimental suggestion would substantially strengthen the paper if the strains were easily available for testing. However if this study would substantially delay the publication of this work, it was not deemed essential. "It is interesting that eif-3.g(C130Y) decreases translation of hlh-30 and ncs-2 only in acr-2(gf) mutants. Is this a direct consequence of elevated neural activity in Ach MNs? For example, does silencing of these neurons attenuate the ability of eif-3.g(C130Y) to decrease ncs-2 GFP reporter translation?"

We thank the reviewers for raising this point. We have examined HLH-30::GFP reporter translation in an *unc-13*(*0*) background, which is deficient for the synaptic priming factor and severely impairs synaptic transmission in all neurons (Richmond *et al.*, 1999) and abolishes *acr-2*(*gf*) convulsion behavior (Zhou *et al.,* 2013). We found that *unc-13*(*0*) prevents enhanced HLH-30 expression in *acr-2*(*gf*) single mutants and also attenuates the ability of *eif-3.G*(*C130Y*) to decrease HLH-30 expression in *acr2*(*gf*). This new data suggests that the observed *eif-3.G(C130Y)* is specific for elevated neuronal activity in ACh MNs. This analysis is now reported in Figure 6C and lines 347350 of the revised text.

5) The reviewers had many questions about the ife-1 section of the study and felt that it was underdeveloped with respect to the rest of the paper. They suggest the authors consider removing this work from the current study and develop it further for a future publication.

We agree with reviewers’ suggestion, and have removed this data from the revised manuscript.

Reviewer #1:[…] Overall, I found the quality of the science and the data presented to be very high. I have only two primary suggestions for experiments that I feel would strengthen the conclusions of the current manuscript.1) The ∆RRM construct, which is a deletion, may have secondary effects on folding/stability of the encoded protein. Given that this EIF-3.G variant is used as an important control for seCLIP studies, I think it would be useful to check that this RRM deletion mutant protein is at least expressed at similar levels as the EIF-3.G wild type and C130Y variants. Since the authors already have FLAG-tagged versions of all three proteins for their CLIP experiments, it should not be too much work to perform a Western blot to compare protein levels.

We thank the reviewer for suggesting this important control. As responded to essential point 1: we have performed western blots as suggested. The results are shown in (Figure 4—figure supplement 1A). While we found that by western blotting the 3XFLAG::EIF-3.G(∆RRM) transgene is expressed at reduced levels compared to that of the EIF-3.G(WT) and EIF-3.G(C130Y) transgenes, we immunoprecipitated EIF3.G(∆RRM) at roughly equivalent levels to EIF-3.G(WT) and EIF-3.G(C130Y) during seCLIP due to the saturating effect of overnight immunoprecipitation. Furthermore, replicate seCLIP experiments using the EIF-3.G(∆RRM) transgene generated roughly three times the number of reads compared to seCLIP from the no transgene control (Supplementary File 4). We hope the reviewer agrees that the EIF-3.G(∆RRM) transgene serves as an effective control in our seCLIP analysis to identify binding sites specific to the RRM of EIF-3.G.

We describe the results in the revised results (lines 220-226) and methods (lines 652-661).

2) The connection to structured 5'UTRs is intriguing but could be further strengthened by an experiment that more explicitly tests the role of secondary structure. For example, in addition to the UTR swapping experiments performed by the authors, it would also be informative to introduce mutations at nucleotides engaged in base-pairing interactions to see if eliminating these interactions influences regulation. If effects are observed, subsequent compensatory mutations that re-introduce base-pairing should be performed to revert back to patterns observed with the native 5'UTR.

We agree with the reviewer on the idea to strengthen the connection of structured 5’UTRs in regulating translation levels by *eif-3.G*(*C130Y*). We selected the *ncs-2* 5’UTR to test the idea. Using RNAfold, we identified a potential stem-loop/hairpin near the initiation codon ATG. While we used mutagenesis to alter this potential structure, after analyzing a number of transgenic lines, we encountered an unexpected outcome that the mutated 5’UTR led to mis-expression in other tissues. We speculate that these nucleotides may critically contribute to tissue-specific gene expression patterns in addition to the regulation of NCS-2 translation via EIF-3.G. Therefore, we did not further pursue this approach, although we will continue to investigate mechanistic basis of the 5’UTR-mediated regulation by EIF-3.G in our future studies.

Reviewer #2:[…] 1) The study provides a characterization of a new eif-3.g allele, which is interesting. However, the study does not come full circle and clarify how the eif-3.g mutation alters cholinergic neuron excitability. Nor does it show in a mechanistic sense how eif-3.g function depends on neuronal activity. It would be helpful if the authors could point to any efforts that they've made to revert the phenotype of eif-3.g via overexpression of eif-3.g(c130y)-downregulated target genes. If they were able to show such an effect, this would greatly enhance the study.

We thank the reviewer for this suggestion and agree that showing such an effect would enhance our study. We generated strains overexpressing transgenes of the EIF3.G target genes *ncs-2*, *flp-18*, and *sbt-1*, but none of the transgenes reverted convulsion behavior in *eif-3.G*(*C130Y*); *acr-2*(*gf*) double mutants. These observations are in-line with our previous and recent publications (Stawicki et al., 2013, and McCulloch et al., 2020) showing that it takes co-expression of multiple insulin genes or multiple *flp* and *nlp* genes to affect *acr-2*(*gf*). Our data supports *eif-3.G*(*C130Y*) modulation of convulsion behavior as the result of altered expression of many genes that together affect ACh-MN activity.

2) It is interesting that eif-3.g(C130Y) decreases translation of hlh-30 and ncs-2 only in acr-2(gf) mutants. Is this a direct consequence of elevated neural activity in Ach MNs? For example, does silencing of these neurons attenuate the ability of eif-3.g(C130Y) to decrease ncs-2 GFP reporter translation?

We appreciate the reviewer’s suggestion for testing this model. As responded above in essential point 4: We found that *unc-13*(*0*) prevents HLH-30 upregulation in *acr-2*(*gf*) and attenuates the ability of *eif-3.G*(*C130Y*) to decrease HLH-30 expression in *acr-2*(*gf*). The new data suggests that the observed *eif-3.G(C130Y)* is specific for elevated neuronal activity in ACh MNs, and is shown in Figure 6C and described in the revised text (lines 347-350).

3) A couple of extra genetic controls would bolster the paper: (i) using CRISPR to engineer a revertant of the C130Y eif-3.g mutation and asking whether this abolishes the phenotype; and (ii) overexpressing eif-3.g(C130Y) in Ach MNs and asking whether this is sufficient to induce the phenotype.

We thank the reviewer for proposing these controls. As responded above in essential point 2: We took the reviewers suggestion (point ii) and expressed the GFP::EIF-3.G(C130Y) transgene in the *eif-3.G*(*0*) background, which is deficient of endogenous EIF-3.G, and found that it strongly suppresses *acr-2*(*gf*) convulsions. This result is reported in Figure 2A and described in the revised text (lines 158-163).

In addition, we used CRISPR editing to change the first cysteine of the EIF-3.G zinc finger CCHC motif to tyrosine (C127Y) and found that it affects *acr-2*(*gf*) behavior to the same extent as *eif-3.G*(*C130Y*), suggesting *acr-2*(*gf*) behavior is generally affected by altered EIF-3.G zinc finger function. The new result is reported in Figure 1C and in the revised manuscript (lines 107-113, 432-434, and 528-535).

Together, these results further support the conclusion that *eif-3.G*(*C130Y*) is the mutation causing suppression of *acr-2*(*gf*).

4) The use of neuronal cell type-specific seCLIP is exciting, but only 2 replicates were conducted and there is no systematic validation of the detected binding sites. RT-qPCR should be used to validate detected binding sites on a handful of targets.

We agree that a validation of seCLIP targets using an RT-qPCR approach would improve confidence in our seCLIP results. However, as most of the targets show broad expression in multiple tissues, including intestine, the RNA isolated from the entire worm have very low or little representation of neuronal transcripts. This is why we chose to examine GFP reporter expression for EIF-3.G target genes. As responded to the essential point 3, we screened ten reporters of EIF-3.G target genes reporters (including *hlh-30* and *ncs-2*), and found six showing detectable expression in ACh-MNs. As with NCS-2 and HLH-30, we found that ATF-7 and LET-607 expression in the ACh-MNs is reduced specifically in the *eif-3.G*(*C130Y*); *acr-2*(*gf*) background. We have added our observations of these reporters in the revised Supplementary File 1.

5) For Figure 4E-F and related analyses: the authors are comparing the eif-3.g targets to the full transcriptome to determine notable properties of the eif-3.g-bound gene set. I believe that for these analyses it would be more appropriate to compare the eif-3.g-bound gene set to the set of ACH MN-expressed genes (which the authors have measured in a previous study). In its current form, it is unclear whether the length and GC content of the eif-3.g-bound genes is a general property of cholinergic neuron gene expression. The same idea also applies to the GO analyses.

We thank the reviewer for this suggestion. In our revised Figures 4E-F, we now show that the length and GC-content of 5’UTRs among EIF-3.G targets (n=179) is significantly longer than that of the cholinergic transcriptome (n= 4,573; McCulloch et al., 2020). The results are described in the revised text (lines 270-278).

6) Related to the length of the eif-3.g-bound 5'UTRs, longer RNA sequences have a higher probability of having a detected binding site (i.e. the number of detected binding sites along any region of DNA/RNA should scale with length). It is not clear whether the authors correct for this in their analysis.

The reviewer raises a valid concern that the number of seCLIP detected binding sites should scale with transcript length. Our initial detection of read clusters comprising EIF-3.G footprints in 5’UTRs was independent of their length. This is because we grouped all clusters located in either the CDS or 5’UTR into one category (5’UTR proximal), since CDS clusters in our seCLIP data were almost always located within 200nts of a 5’UTR (Figure 4D). We now clearly state this detail in the revised methods (lines 717-719).

It is also important to note that we identified one cluster per gene on average for each EIF-3.G transgene (Supplementary File 5). Furthermore, we identified EIF-3.G binding sites in many small transcripts such as neuropeptide genes, which are among the shortest transcripts in the *C. elegans* transcriptome and were among the most highly enriched in our dataset of EIF-3.G targets (Figure 5A). Together, these data suggest that EIF-3.G binding sites are detected independent of transcript length.

Reviewer #3:[…] 1. The statistical representation in Figure 1E is somewhat misguided. In order to show that eif3.G increases convulsion in acr-2 (gf); eif-3.G(C130Y) in an ACh-MN-specific manner, the statistical analysis of the right 4 samples should be done with WT or (-), not with Ex[eif-3.G(+)].

Thank you for pointing us to this error. We have now corrected Figure 1E with statistical analysis against the WT control instead of with Ex[*eif-3.G*(+)].

2. The statement 'Both GFP::EIF-3.G(WT) and EIF-3G(C130Y) showed cytoplasmic fluorescence throughout somatic cells and across all developmental stages' is not well-presented in Figure 1—figure supplement 1B, since only the wild-type and one developmental stage (L4) was shown. It is also unclear if the bright puncta across the animal actually correspond to Peif-3.g::GFP or is a result of autofluorescence from the gut. It will be useful to include arrows and insets to highlight the pattern of interest (e.g., cytoplasmic, specific somatic tissues). To elaborate on the tissue specificity suggested in Figure 1E, whether EIF-3.G WT and C130Y are expressed in ACh-MN neuron should be discussed – This will differentiate whether the C130Y effect is cell-autonomous or non-autonomous. The authors also referred to Figure supplement 1B and Figure 2B as assessments of EIF-3.G protein stability – Expression pattern/localization would be more appropriate than protein stability here.

We thank the reviewer for pointing out the issue and for suggestions. We have added new images of L4 animals expressing GFP::EIF-3.G(WT) and GFP::EIF-G(C130Y) to the revised manuscript (Figure 1—figure supplement 1B), which shows that both GFP::EIF-3.G(WT) and GFP::EIF-3.G(C130Y) show indistinguishable tissue expression pattern through somatic cells.

We revised our description of the data in the revised text (lines 153-155) to

“Fluorescence from both GFP::EIF-3.G(WT) and GFP::EIF-3.G(C130Y) was observed

in all somatic cells (Figure 1—figure supplement 1B). In ACh-MNs, both proteins showed cytoplasmic localization (Figure 2B).”

We also edited the Figure 1—figure supplement 1B legend in the revised text (lines 958-962) to make clear that the bright punctae observed across the animal midbody are auto-fluorescent gut granules.

In addition, our conclusions regarding EIF-3.G protein stability now refer exclusively to the analysis in Figure 2B (lines 163-166 of the revised text).

3. In Figure 3A, the potential involvement of ife-1 (one of the eIF4E homologs) in eif-3.g(C130Y) suppression of acr-2(gf) is interesting. Some experiments are needed to discern the direct/indirect nature of this genetic interaction.A. Does ife-1(0) cause convulsion in eif-3.G(C130Y) alone (acr-2 wild-type)?B. Does ife-1(0) alter eif-3.G WT and C130Y expression?C. ife-1 was shown to be predominantly expressed in the germline where it is important for germ cell development (Henderson et al., Journal of Cell Science 2009; Amiri et al., Development 2001). Moreover, the protein is undetectable by western blot from germline-deficient glp-4(bn2) animal (Amiri et al. 2001), potentially indicating exclusive germline expression. Thus, it will be important to look at whether IFE-1 is co-expressed with EIF-3.G WT and C130Y in the soma and ACh-MN.

We thank the reviewer for posing these questions. As recommended in the editorial summary, we have removed data pertaining to *ife* genes. We will keep reviewer’s questions in mind in our future analyses of these genes.

4. EIF-3.G preference for long 5' UTR with high GC content was nicely dissected from the seCLIP data in Figure 4. For consistency, the authors could provide information of 5'UTR length and GC content between upregulated and downregulated genes in acr-2(gf) over WT, shown in Figure 5B. hlh-30 and ncs-2 were further linked to eif-3.g-mediated suppression of acr-2(gf) by examining their expression using GFP reporters in Figures 6 and 7. Polysome profiling can be done to differentiate the effect of C130Y on the translation efficiency of these transcripts, as opposed to their transcription and protein stability. It would also be informative to show the preference of these genes for *C. elegans* eIF4E homologs, IFE-1, IFE-2, and IFE-4.

We appreciate this suggestion. We performed an analysis of 5’UTR length and GC-content among GO categories as well as up- and down-regulated genes, but did not observe significant correlation between them and therefore did not include this data.

We performed qRT-PCR for *ncs-2* and *hlh-30* transcripts from polysome fractions prepared from whole worm lysates, as the reviewer suggested. Unfortunately, we did not obtain meaningful results from this analysis, partly due to technical limitation as polysome profiles from whole worm lysates lack specificity to the ACh-MNs where *eif3.G*(*C130Y*) acts to suppress *acr-2*(*gf*).

5. A heatmap in Figure 4 —figure supplement 2C needs hierarchical clustering, even though genes in the map are already clustered top to bottom by increasing positional GC-density near the start codon. It may help extract additional meaningful information. The authors also need to point out the genes described in the text (pmt-2, let-607, unc-43, and gsa-1).

We thank the reviewer for suggesting this analysis. We have added hierarchical clustering to the heatmap in Figure 4—figure supplement 2C, which has highlighted interesting examples of GC-density positioning for the genes (*zip-2*, *sec-61*, *pdf-1*, and *kin-10*). We now discuss in the revised text (lines 278-283) and point out genes described in the text within the figure as suggested.

6. The statistical representation in the graph in Figure 6D needs to be corrected.

Thank you! We have corrected the statistical representation in the revised Figure 6D.